# Organic matter composition and greenhouse gas production of thawing subsea permafrost in the Laptev Sea

Birgit Wild [1,2] ✉, Natalia Shakhova[3,4], Oleg Dudarev[3,4], Alexey Ruban[4,5], Denis Kosmach[3,4], Vladimir Tumskoy[4,6], Tommaso Tesi[7], Hanna Grimm[1,8], Inna Nybom [1], Felipe Matsubara[1,2], Helena Alexanderson [9], Martin Jakobsson [2,10], Alexey Mazurov[5], Igor Semiletov[3,4,11] & Örjan Gustafsson [1,2] ✉

Subsea permafrost represents a large carbon pool that might be or become a significant greenhouse gas source. Scarcity of observational data causes large uncertainties. We here use five 21-56 m long subsea permafrost cores from the Laptev Sea to constrain organic carbon (OC) storage and sources, degradation state and potential greenhouse gas production upon thaw. Grain sizes, optically-stimulated luminescence and biomarkers suggest deposition of aeolian silt and fluvial sand over 160 000 years, with dominant fluvial/alluvial deposition of forest- and tundra-derived organic matter. We estimate an annual thaw rate of $1.3 \pm 0.6$ kg OC m$^{-2}$ in subsea permafrost in the area, nine-fold exceeding organic carbon thaw rates for terrestrial permafrost. During 20-month incubations, CH$_4$ and CO$_2$ production averaged 1.7 nmol and 2.4 μmol g$^{-1}$ OC d$^{-1}$, providing a baseline to assess the contribution of subsea permafrost to the high CH$_4$ fluxes and strong ocean acidification observed in the region.

Subsea permafrost represents a large and potentially vulnerable organic carbon pool, yet also one of the least constrained compartments of the cryosphere-carbon-climate system. Subsea permafrost might extend up to $2.5 \times 10^6$ km$^2$ across the Arctic Ocean shelf seas[1]. The majority ($1.4 \times 10^6$ km$^2$) lies underneath the East Siberian Arctic Shelf (ESAS), the World's largest and shallowest continental shelf sea that comprises the Laptev, East Siberian, and Russian Chukchi Seas (Fig. 1). Today's subsea permafrost formed during the Pleistocene when sea levels were lower and the ESAS was part of Beringia, a continuous land mass extending from eastern Siberia over Alaska to western Canada. Beringia was largely non-glaciated during the Last

Glacial Maximum and accumulated thick permafrost deposits during the late Pleistocene. These include Ice Complex deposits (ICD; also known as Yedoma) that show high ice and organic carbon content compared to other mineral permafrost types, but also fluvial/alluvial deposits and thermokarst deposits formed during warmer periods[2]. Part of this permafrost is still preserved on land; another part was eroded by rapid sea level rise after the Last Glacial Maximum or inundated as subsea permafrost[3–5]. It is unclear what fraction of the original permafrost is still preserved underneath the ESAS. The stratigraphy of permafrost deposits along current coastlines however suggests that ICD on today's ESAS were largely destroyed by erosion and

[1]Department of Environmental Science, Stockholm University, 11418 Stockholm, Sweden. [2]Bolin Centre for Climate Research, Stockholm University, 11418 Stockholm, Sweden. [3]Il'ichov Pacific Oceanological Institute, Far-East Branch of the Russian Academy of Sciences, Vladivostok 690041, Russia. [4]Tomsk State University, Tomsk 634050, Russia. [5]Tomsk Polytechnic University, Tomsk 634050, Russia. [6]Melnikov Permafrost Institute, Siberian Branch of the Russian Academy of Sciences, Yakutsk 677010, Russia. [7]Institute of Polar Sciences, National Research Council, 40129 Bologna, Italy. [8]Geomicrobiology, Department of Geosciences, University of Tuebingen, 72076 Tuebingen, Germany. [9]Department of Geology, Lund University, 22362 Lund, Sweden. [10]Department of Geological Sciences, Stockholm University, 11418 Stockholm, Sweden. [11]Institute of Ecology, Higher School of Economics, Moscow 101000, Russia. ✉e-mail: birgit.wild@aces.su.se; orjan.gustafsson@aces.su.se

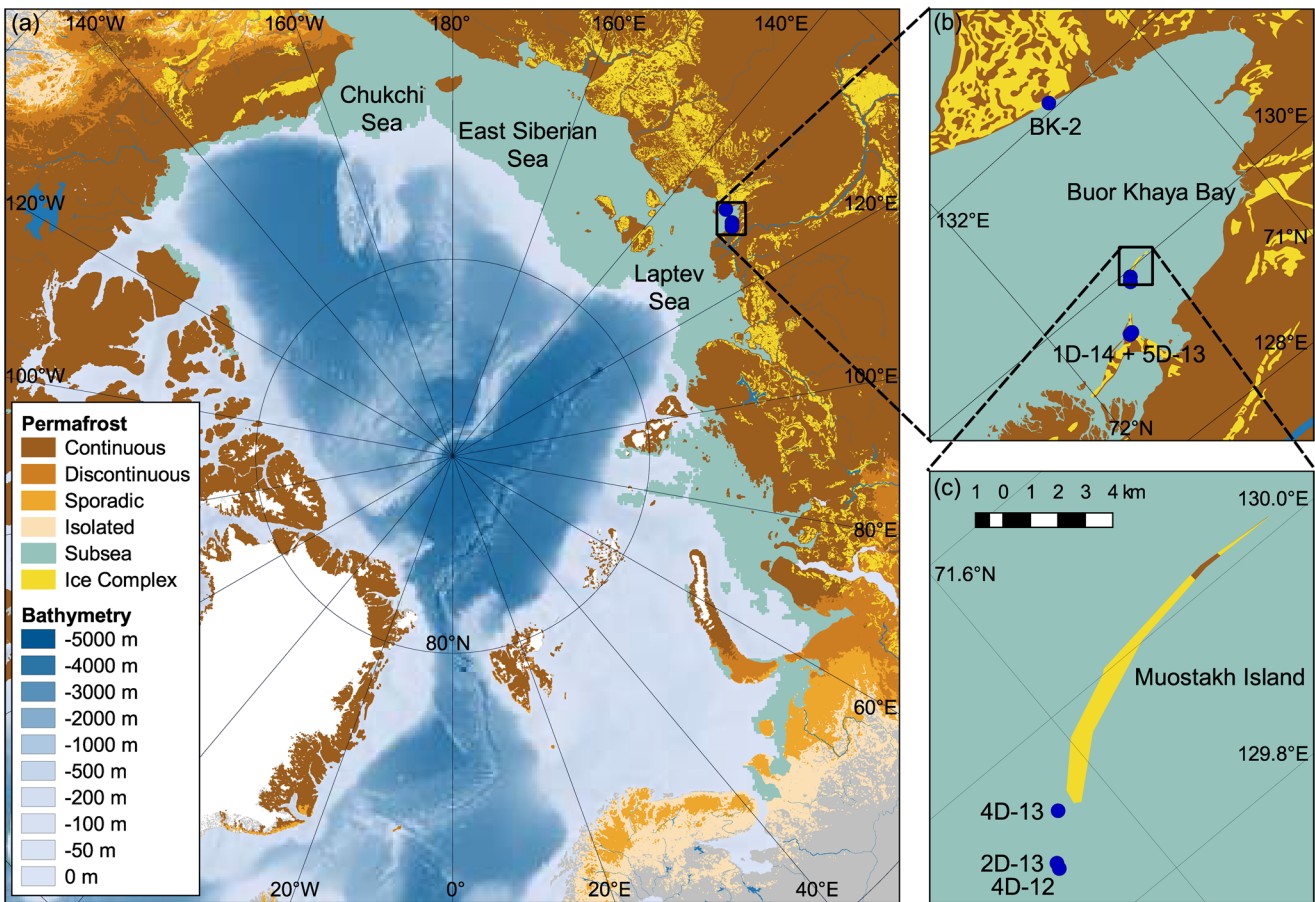

**Fig. 1 | Arctic permafrost extent. a** Shown are subsea[1] and terrestrial[69] permafrost, including Ice Complex deposits[70]. Detailed maps show **b** the Buor-Khaya Bay, with subsea permafrost drill sites described in this (4D-13, 2D-13, 4D-12, 1D-14, 5D-13) and a previous study (BK-2)[28], and even more detailed, **c** Muostakh Island, with subsea permafrost drill sites of cores 4D-13, 2D-13, and 4D-12.

that subsea permafrost represents deeper and older deposits[3,6]. Extensive coastal erosion is still ongoing along the ESAS at rates of up to 5 m per year[7], leading to the inundation, transfer, and potentially mineralization, of substantial amounts of old carbon. This process is likely accelerating with recent warming[8].

Permafrost deposits store large amounts of organic matter that is protected from microbial decomposition while frozen[9]. When thawed, this organic matter can be converted to greenhouse gases such as $CO_2$ and $CH_4$[10,11] and further accelerate global warming: the permafrost-carbon-climate feedback. Terrestrial permafrost has been extensively studied over the past decades, resulting in great advances in our understanding of the stocks[9], quality[12–14], and mineralization of organic carbon to $CO_2$ and $CH_4$[11]. Subsea permafrost is more difficult to access, observational data are scarce, and even many basic properties are completely unknown or poorly constrained. These include the amount and quality of the subsea permafrost organic carbon stock, its vulnerability to mineralization upon thaw, and the resulting potential for greenhouse gas emissions to the atmosphere[15,16].

At the same time, subsea permafrost is warming and thawing more rapidly than its terrestrial counterpart. Subsea permafrost on the ESAS is likely warmed from above by overlying sea water[16,17], and from below by geothermal heat flow[17,18]. In the Buor-Khaya Bay in the southeastern Laptev Sea (Fig. 1), subsea permafrost is warmer than nearby terrestrial permafrost, with subsea permafrost temperatures close to 0 °C to a depth of dozens of meters[19]. Repeated drilling of subsea permafrost in 1982/1983 and 2013/2014 gave the unique possibility to compare the ice-bonded permafrost table (IBPT). This comparison indicated high rates of thaw, with IBPT deepening by on average

14 ± 3 cm year$^{-1}$ in this area[16]. This rate exceeds 35-fold that of terrestrial permafrost in central and eastern Siberia (average 0.4 cm year$^{-1}$, 1990–2020[20,21]). Direct measurements of IBPT deepening are not available for other subsea permafrost areas. Subsea permafrost warming and thawing are expected to continue and possibly intensify in the future if water temperatures continue to rise with anthropogenic climate warming.

Organic matter degradation upon subsea permafrost thaw is a possible source of $CH_4$ and $CO_2$ to ESAS waters and the atmosphere. High variability in $CO_2$ and $CH_4$ production has been described for terrestrial permafrost[11] and might be related to differences in paleoenvironment during formation, organic matter composition, and microbial community composition[22–25]. The potential for $CO_2$ and $CH_4$ production by organic matter degradation in subsea permafrost after thaw is unknown. Field campaigns to the ESAS have observed strongly elevated $CH_4$ concentration in seawater above subsea permafrost compared to the atmospheric equilibrium[19,26,27]. A current key challenge is to constrain the contribution of $CH_4$ from different possible sources[26–29]. These include the microbial decomposition of thawed subsea permafrost organic matter, but also the release of preformed $CH_4$ stored in subsea permafrost, in shallow $CH_4$ hydrates[30], and from deeper, thermogenic $CH_4$ pools[27,31]. The hydrate and thermogenic $CH_4$ might be trapped within and underneath frozen subsea permafrost and may escape to the surface as permafrost thaws and gas migration pathways form[16,26,30]. The ESAS also shows particularly high $CO_2$ concentration and ocean acidification, possibly linked to $CO_2$ production during the decomposition of organic matter transported from land[32]. The large extent of subsea permafrost on the Arctic Ocean shelves and

its potential as a $CO_2$ and $CH_4$ source highlight the need for observational data on subsea permafrost, including organic matter composition and vulnerability to decomposition upon thaw. At present, subsea permafrost remains one of the largest uncertainties concerning future Arctic carbon dynamics and greenhouse gas emissions.

This study aims to improve the understanding of the subsea permafrost organic matter pool and assess the potential for greenhouse gas production by recommencing decomposition after the recent thaw. To this end, we used a unique set of five 21–56 m long cores that were drilled into the subsea permafrost from the sea ice in the Buor-Khaya Bay in 2012–2014 to target four objectives.

(1) The first objective was to constrain organic matter sources at the current thaw front and test the hypothesis that subsea permafrost does not represent ICD but deeper deposits. Understanding the nature of subsea permafrost is a prerequisite for reconstructing permafrost destruction and organic matter mobilization during past periods of rapid warming and sea level rise, and thereby inform projections of permafrost dynamics under ongoing climate change. Information on organic matter sources also helps to assess its vulnerability to decomposition by comparison with analogous terrestrial deposits where data are more abundant. Optically stimulated luminescence (OSL) dating of one core together with previously published grain size data[16,33] was employed to place the cores in the stratigraphic context of the study region. We analyzed material from five depths of each core for organic carbon content, and then focused on 3 m increments above and below the IBPT of three cores where we performed high-resolution analysis of organic carbon content and isotopic composition, as well as a suite of lignin and lipid biomarker proxies that inform about organic matter sources.

(2) The second objective was to quantify organic carbon storage and thaw based on observed organic carbon concentrations, to provide quantitative constraints for modeling efforts.

(3) Third, we assessed the degradation state of organic matter at the current thaw front, hypothesizing that recommencing degradation after thaw would be reflected in changing molecular properties of lignin and lipid compound classes.

(4) We finally quantified potential rates of $CH_4$, $CO_2$, and $N_2O$ production by organic matter decomposition after thaw, in a 20-month incubation experiment of selected samples. Taken together, this study contributes to our understanding of the subsea permafrost organic matter pool and its potential as a source of greenhouse gases by decomposition after thaw.

## Results and discussion

### Stratigraphic context

The marine transgression starting at the end of the last glacial period flooded a complex permafrost landscape with intact and thermokarst-affected ICD, lakes, and rivers[3]. Terrestrial sites near the Buor-Khaya Bay show a sequence of deposits, with Holocene deposits at the top that are frequently underlain by ICD (10–50 ka)[2], but also fluvial/alluvial sands from the MIS 4/5a-d (50–110 ka), thermokarst deposits formed during the MIS 5e interglacial (115–130 ka), and older ICD from the MIS 6/7 stadial (130–200 ka)[2,6].

Optically stimulated luminescence dating together with grain sizes are consistent with the hypothesis that subsea permafrost of the Buor-Khaya Bay largely does not represent ICD but instead older material of different origin. Depositional ages of core 4D-12 fall between 162 ka at 51 m depth and 51 ka at 17 m depth below the sea floor (Fig. 2 and Supplementary Table 1). Although this period covers a considerable range in climatic conditions, a close, linear correlation of age and depth (Pearson's correlation, $R^2 = 0.98$, $n = 5$) suggests a rather constant deposition at least at the low temporal resolution available. The measured age of $8.5 \pm 0.6$ ka at 15 m depth falls outside this pattern and reflects potentially a period of low deposition or an erosional

event. Even excluding this single observation, the comparatively young OSL ages observed here at great depths of core 4D-12 contrast to previously determined radiocarbon ages of >40 ka in ICD above sea level around the Buor-Khaya Bay, including on Muostakh Island[34,35]. This could indicate inconsistencies in OSL vs radiocarbon-based ages, or a large shift in depositional environment between coastal and subsea permafrost sediment sequences.

The subsea permafrost cores were characterized by high variability in grain size distribution. Cores 4D-12 and 2D-13 showed alternations between finer (silt, clay) and coarser-grained (sand) deposits with mostly unimodal grain size distributions (Fig. 2). These grain size distributions contrast to the typically multimodal distribution of ICD with peaks in both silt and fine sand fractions[35,36], and rather suggest fluctuations between predominant wind- and water-related deposition. Core 4D-13 showed largely bimodal distributions with silt and sand at the same depth. Predominantly sandy material has been previously described for core BK-2 from the eastern Buor-Khaya Bay[28] where subsea permafrost also developed by coastal erosion and inundation as for our locations near Muostakh Island. In Ivashkina Lagoon where subsea permafrost was formed by salinization of a coastal thermokarst lake, mostly sandy material was observed for 1D-14, and a transition from silt to sand with depth for 5D-13. Overall, pronounced differences in grain size distribution point at high spatial variability of deposition regimes even within the small study area of the Buor-Khaya Bay.

### Organic matter sources

Organic matter properties at the current thaw front of subsea permafrost were compared with previously published data from terrestrial deposits in northeast Siberia that could resemble the original state of subsea permafrost before inundation. These include (1) Pleistocene ICD, (2) Pleistocene and Holocene thermokarst deposits, (3) Holocene peat permafrost, and (4) Holocene active layers. We further included translocated organic matter in our comparison, such as (5) Pleistocene fluvial/alluvial deposits, (6) contemporary river-suspended material where data from fluvial/alluvial deposits were not available, and (6) contemporary, marine surface sediments from the Buor-Khaya Bay that receives strong input from the Lena River.

Predominant fluvial/alluvial deposition of organic matter at the thaw front of subsea permafrost near Muostakh Island was indicated by the content and $^{13}C$ isotopic composition of organic carbon and a suite of biomarker source proxies. Organic carbon contents averaged $0.8 \pm 0.3\%$ (mean ± standard deviation) for 4D-13, 2D-13, and 4D-12, as well as $0.5 \pm 0.7\%$ as previously described for core BK-2[28]. Organic carbon contents showed no systematic trends across these cores and fell in the same range as those of fluvial/alluvial sediments deposited during different periods of the Pleistocene[2] (Fig. 3). Organic carbon contents of subsea permafrost were in general lower than in Pleistocene ICD, thermokarst deposits[2], and near-by Buor-Khaya Bay surface sediments[14,37]. Cores 5D-13 and 1D-14 from Ivashkina Lagoon showed higher organic carbon contents at the surface (Supplementary Table 2) that likely reflect Holocene sediments, with an overall average of $1.4 \pm 1.4\%$ (previous[38] plus own data; one observation of 26% excluded as an outlier). The $^{13}C$ isotopic composition of organic carbon provides additional information on organic carbon sources. Eastern Siberia is dominated by C3 vegetation with $\delta^{13}C$ values commonly between −25‰ and −30‰[39]. Terrestrial permafrost in the region, such as ICD and thermokarst deposits, fall at the lower end of this range[2], and fluvially or marine translocated deposits at the upper end[2,37] (Fig. 3). This gradient reflects changes in the isotopic composition of organic matter by processing during transport, together with mixing with comparatively $^{13}C$-enriched organic matter from aquatic primary production. Average $\delta^{13}C$ values of 4D-13, 2D-13, and 4D-12 were in line with organic matter modification during fluvial translocation (−25.0 ± 1.0‰). Cores 5D-13 and 1D-14 from Ivashkina

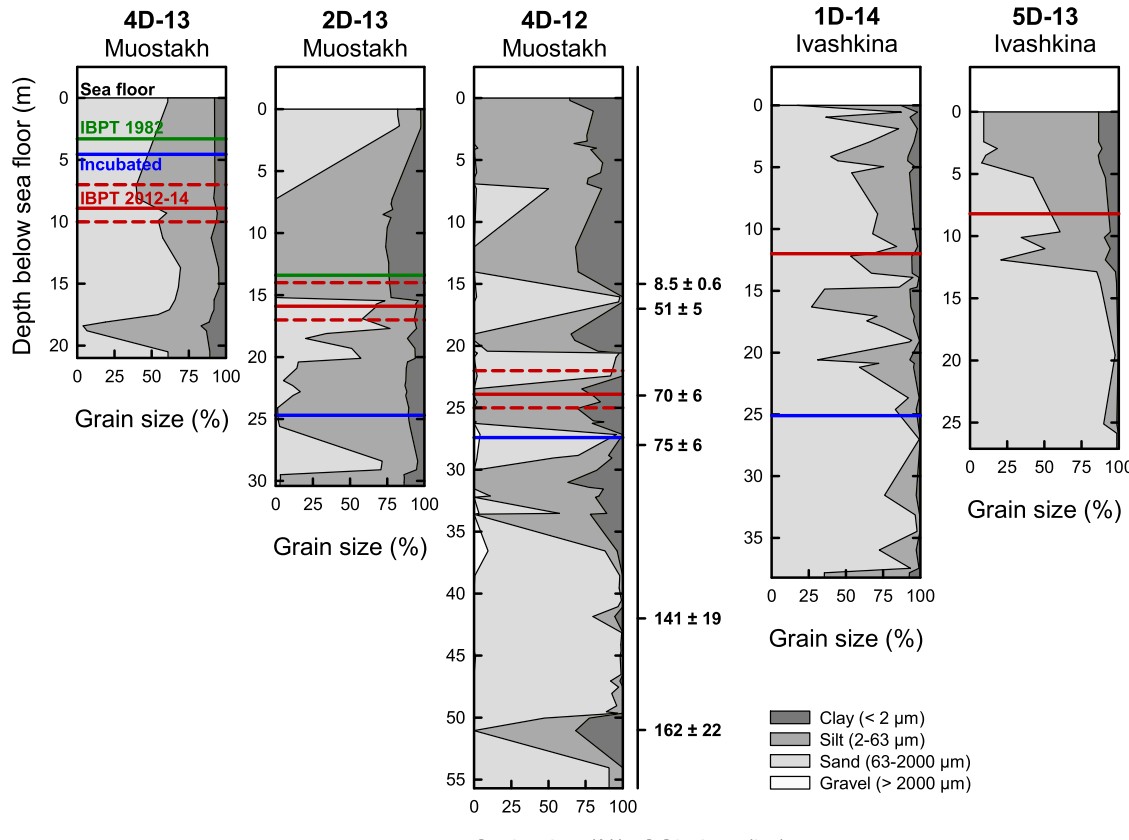

**Fig. 2 | Grain size distribution and optically stimulated luminescence (OSL) ages of subsea permafrost cores.** Cores were drilled near Muostakh Island and in the Ivashkina Lagoon in the Buor-Khaya Bay. The position of the ice-bonded permafrost table (IBPT) in 2012−2014 and 1982 is indicated by red and green lines, respectively. The interval marked between dashed red lines shows the thaw front

sections where detailed organic matter analyses have been performed. The blue lines indicate samples used in the incubation experiment. Grain size and IBPT data (except 5D-13) are from previous publications[16,33]. Detailed grain size data are provided in Supplementary Table 8.

Lagoon showed more depleted values at the surface leading to higher down-core variability (−25.4 ± 1.7‰), and core BK-2 had overall more depleted $\delta^{13}C$ values of −26.3 ± 0.8‰ as previously described[40]. Similarly, mass ratios of total organic carbon over total nitrogen (OC/TN) were higher at the surface and more variable in 5D-13 and 1D-14 (15.1 ± 4.8) compared to 4D-13, 2D-12, and 4D-12 (9.4 ± 2.5; Fig. 3).

Biomarker proxies based on $n$-alkanes and lignin phenols permit a more detailed assessment of organic matter sources to subsea permafrost near Muostakh Island. These support contributions from boreal forests, tundra and—to a small extent—peatlands, as well as the modification of terrigenous organic matter during aquatic transport. Concentrations of terrestrial high-molecular-weight (HMW) $n$-alkanes, $n$-alkanoic acids and $n$-alkanols, as well as lignin phenols amounted to 2.7 ± 2.6, 1.5 ± 1.8, 1.0 ± 0.5, and 5.3 ± 5.2 mg g$^{-1}$ OC, respectively (Supplementary Fig. 1).

The modification of terrigenous organic matter during aquatic transport is supported by terrigenous-to-aquatic ratios (TAR), complementing organic carbon δ$^{13}$C values. The TAR builds on the much higher abundance of HMW relative to low-molecular-weight (LMW) $n$-alkanes in higher plants compared to lower plants such as algae and mosses[41]. In northeast Siberia, TAR sharply delineates terrestrial vs aquatic deposits following the abundance of higher vs lower plants in these systems (Fig. 4). Ice Complex and thermokarst deposits showed much higher TAR than Buor-Khaya Bay sediments that however still by far exceeded 1 (19 ± 3)[42]. This indicates a higher contribution of $n$-alkanes from lower plants than in ICD and thermokarst deposits, but

still a dominance of higher-plant $n$-alkanes due to high input of land-derived organic matter by the Lena River. The TAR values of subsea permafrost near Muostakh Island fall in the same range and are in line with the TAR signal of terrigenous organic matter after aquatic translocation, rather than with that of the terrigenous source itself. An even weaker terrestrial signal has been previously reported for subsea permafrost in Ivashkina Lagoon[43].

Lignin proxies and $C_{25}/(C_{25} + C_{29})$ $n$-alkane ratios allow to assess the contribution of organic matter from different terrestrial sources. Low $C_{25}/(C_{25} + C_{29})$ $n$-alkane ratios of subsea permafrost indicate a low contribution of peat-forming *Sphagnum* moss. The range observed in subsea permafrost was similar to that of mineral deposits in Siberia, including non-peat active layer[44], ICD[36,44], and Holocene thermokarst[36] (Fig. 4). By comparison, active layer and permafrost material from Siberian peatlands was characterized by higher $C_{25}/(C_{25} + C_{29})$ ratios[45]. Intermediate ratios were observed in contemporary material influenced by aquatical translocation such as Buor-Khaya sediments and suspended material in the Kolyma river, indicating varying contribution of peat material from the respective drainage basins[42,46,47].

Lignin proxies allow to apportion organic matter from different terrestrial higher plants. Syringyl/vanillyl (S/V) and cinnamyl/vanillyl (C/V) lignin phenol ratios reflect the relative contribution of lignin from angiosperm vs gymnosperm and non-woody vs woody plant tissues. The wide S/V and C/V range of 0.59 ± 0.18 and 0.34 ± 0.25 at the thaw front of 4D-13, 2D-13, and 4D-12 indicates a contribution of both woody gymnosperm and non-woody angiosperm tissues (Fig. 5), with

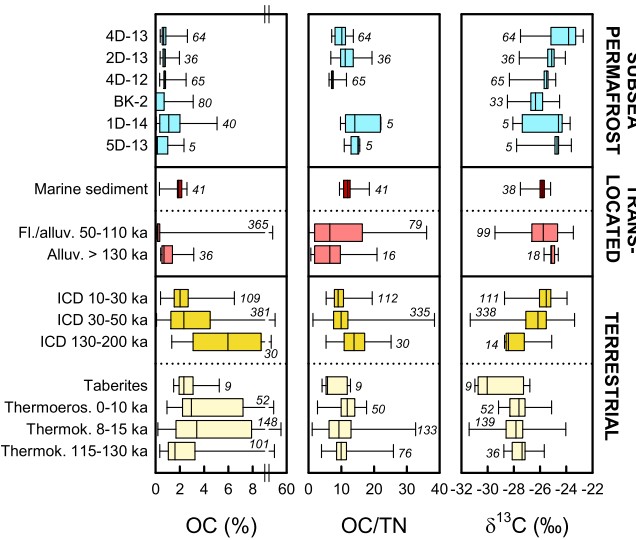

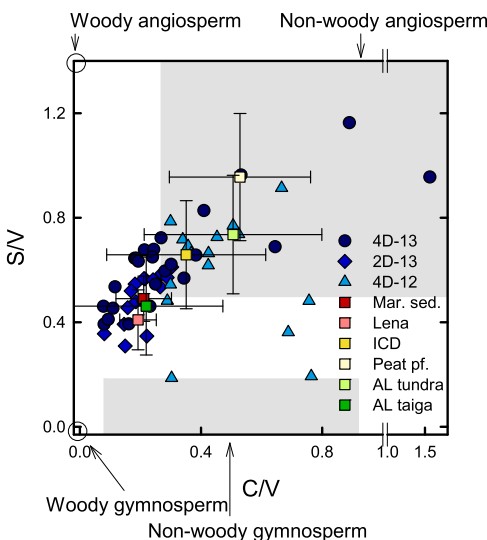

**Fig. 3 | Comparison of organic carbon in subsea permafrost in the Buor-Khaya Bay with other deposits in northeast Siberia.** Shown are organic carbon (OC) content, total organic carbon over total nitrogen (OC/TN), and organic carbon $\delta^{13}C$ values. All data from cores 4D-13, 2D-13, 4D-12, 5D-13, and on OC/TN and $\delta^{13}C$ from core 1D-14 are from this study. Data from previous studies include OC in 1D-14[38] and BK-2[28,40], terrestrial deposits such as Ice Complex deposits (ICD), taberites, thermoerosion (thermoeros.), and thermokarst/lake deposits (thermok.) of different deposition age[2], as well as Buor-Khaya Bay surface sediments[14,37], fluvial and alluvial (fl., alluv.) deposits that are affected by translocation. Box plots show medians with 25th and 75th percentiles as box limits, minimum/maximum values as whiskers, and numbers of observations in italics.

**Fig. 5 | Lignin source proxies in subsea permafrost. Subsea permafrost data represent the thaw fronts of cores 4D-13, 2D-13, and 4D-12 from the Buor-Khaya Bay.** S/V syringyl/vanillyl, C/V cinnamyl/vanillyl. Gray areas indicate ranges of different plant tissue types[14,63,71–74]. Data from previous studies in northeast Siberia are shown for comparison (means ± standard deviation), including Buor-Khaya Bay surface sediments (mar. sed.)[14,42,48], Lena River suspended matter, Ice Complex deposits (ICD)[14,48], Holocene peat permafrost (pf.)[14], and active layer (AL) material from tundra and boreal forest (taiga) locations[48].

changes in their relative proportion over time (Supplementary Fig. 1). Non-woody angiosperm vegetation is characteristic for tundra landscapes with abundant shrubs or grasses that dominated late Pleistocene Beringian landscapes as well as today's coastline. This is reflected also in high S/V and C/V ratios in Pleistocene ICD, Holocene peat permafrost, and tundra active layer soils[14,48]. By contrast, woody gymnosperm material was likely translocated by rivers from boreal forests in the south. Vegetation reconstructions suggest that trees have been sparse in the study region since at least the MIS 5e interglacial[49,50] ca. 115 ka before present, compared to an OSL-dated age of 65–72 ka for the section of core 4D-12 that was analyzed for biomarkers. Similar long-range translocation of forest organic matter can be observed under present conditions in eastern Siberia. Organic material suspended in the Lena River and deposited in Buor-Khaya Bay sediments is characterized by a high contribution of woody gymnosperm tissues that has been inferred to reflect transport from boreal forests far south of the Lena delta[14,48] (Fig. 5).

## Organic carbon storage and thaw

Subsea permafrost represents a large organic matter pool that, if thawed and microbially degraded, might be an increasing source of $CO_2$ and $CH_4$ to the atmosphere. The average organic carbon content of subsea permafrost cores 4D-13, 2D-13, 4D-12, 1D-14, 5D-13, and BK-2 amounted to $0.7 \pm 0.3\%$ or $9.3 \pm 3.6\ mg\ cm^{-3}$, based on the dry sediment mass per total volume of cores 2D-13 and 4D-12 ($1.3 \pm 0.2\ g$ dry weight $cm^{-3}$), and excluding the upper-most meter that might represent Holocene sediments. The rates of subsea permafrost thaw of $14 \pm 3\ cm\ year^{-1}$ in the study area[16] thus correspond to the thaw-out of $1.3 \pm 0.6\ kg\ OC\ m^{-2}\ year^{-1}$. By comparison, the gradual thaw of terrestrial permafrost in northeast Siberia by active layer deepening is here estimated to yield $0.14\ kg\ OC\ m^{-2}\ year^{-1}$ (standard deviation $0.32\ kg\ m^{-2}\ year^{-1}$). This estimation is based on the average active layer deepening rate of $0.4\ cm\ year^{-1}$ in central and eastern Siberia between 1990 and 2020 (stations with minimum 4 years of observation)[20,21], the relative distribution of permafrost soil suborders[9], and their average

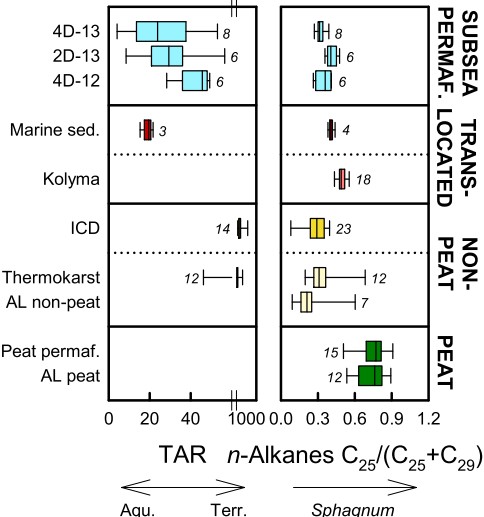

**Fig. 4 | Comparison of *n*-alkane source proxies in subsea permafrost with other deposits in Siberia.** Subsea permafrost data represent the thaw front of cores 4D-13, 2D-13, and 4D-12 from the Buor-Khaya Bay. Shown are terrigenous-aquatic ratios (TAR) reflecting the contribution of aquatic (aqu.) and land-derived (terr.) material, and $C_{25}/(C_{25}+C_{29})$ *n*-alkane ratios reflecting *Sphagnum* contribution. Data from previous studies show terrestrial deposits from peat and non-peat locations including Ice Complex deposits (ICD)[36,44], Holocene thermokarst[36], peat permafrost[45], and active layer (AL) material[44,45], as well as translocated material including Buor-Khaya Bay surface sediments (sed.)[42,46] and Kolyma river suspended matter[47]. Box plots show medians with 25th and 75th percentiles as box limits, minimum/maximum values as whiskers, and numbers of observations in italics.

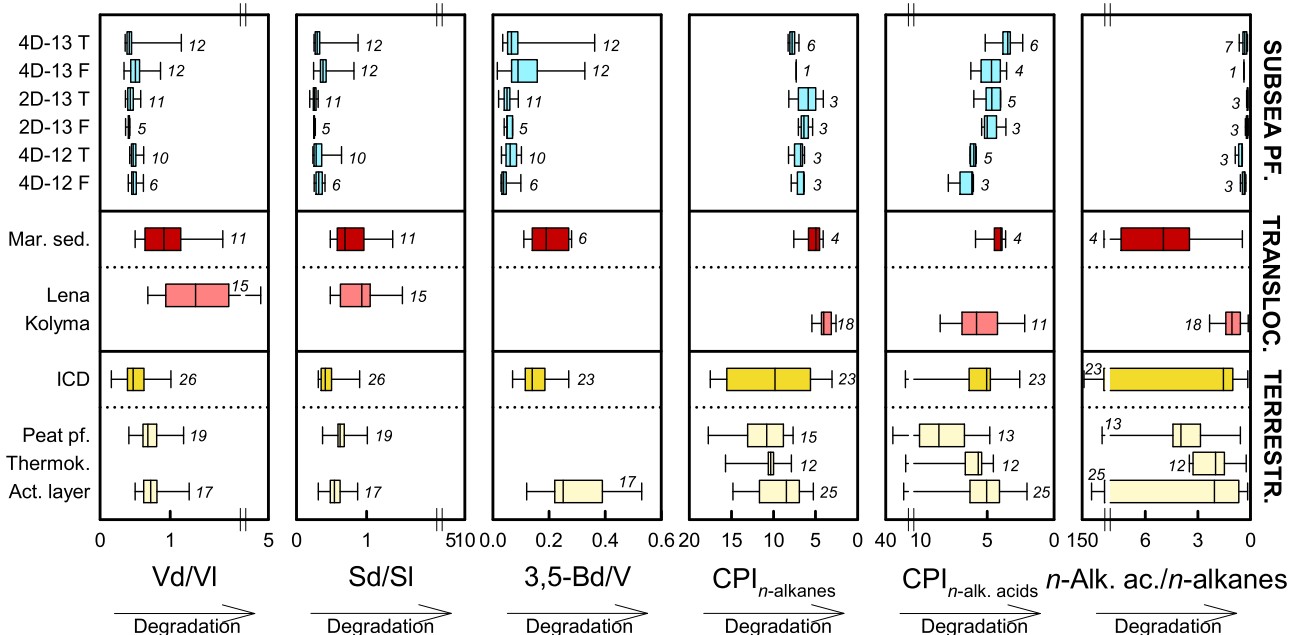

**Fig. 6 | Comparison of biomarker degradation proxies in subsea permafrost with other deposits in northeastern Siberia.** Subsea permafrost (pf.) data represent the thaw fronts of cores 4D-13, 2D-13, and 4D-12 from the Buor-Khaya Bay. Data from previous studies include terrestrial deposits such as Ice Complex deposits (ICD)[14,36,44,48], Holocene peat permafrost[14,45], Holocene thermokarst (thermok.)[36], and active layer material (act. layer)[44,45,48], as well as translocated material such as Buor-Khaya Bay surface sediments (mar. sed.)[14,42,46], Lena[14] and Kolyma[47] river suspended matter. Box plots show medians with 25th and 75th percentiles as box limits, minimum/maximum values as whiskers, and numbers of observations in italics (T: thawed; F: frozen at the time of sampling). Degradation proxies include vanillic acid/vanillin (Vd/Vl), syringic acid/syringaldehyde (Sd/Sl), 3,5-dihydroxybenzoic acid/vanillyl (3,5-Bd/V); carbon preference indices (CPI) of HMW n-alkanes and HMW n-alkanoic acids (n-alk. acids), HMW n-alkanoic acid/ HMW n-alkane ratios.

organic carbon density[51] between the minimum and maximum active layer depth[20,21].

### Degradation state of organic matter at the thaw-front

Organic matter now in subsea permafrost may have been degraded at the site of origin, during transport, after re-deposition before freeze-down, and again after re-thaw. Lignin- and lipid-based degradation proxies suggest that terrestrial organic matter in translocated deposits such as river suspended material and marine sediments[14,42,46,47] is more degraded than in terrestrial (source) deposits[14,36,44,45,48] (Fig. 6). Although bulk organic matter and biomarker source proxies indicate the aquatic translocation and modification of organic matter at the thaw front of subsea permafrost in the Buor-Khaya Bay, differences between biomarker signatures give an inconsistent overall picture of its degradation state. Lignin-based degradation proxies (acid/aldehyde ratios of syringyl and vanillyl phenols, 3,5-dihydroxybenzoic acid/vanillyl ratios) suggest a lower degree of decomposition of organic matter found in subsea permafrost than in terrestrial deposits; lipid-based degradation proxies (carbon preference indices of HMW n-alkanes and HMW n-alkanoic acids, HMW n-alkanoic acid/HMW n-alkane ratios) indicate the opposite (Fig. 6). Correlations among subsea permafrost samples show that strong degradation indicated by lipid-based proxies coincided with low degradation based on lignin-based proxies and vice versa (Supplementary Table 3). It is possible that the observed discrepancies between degradation proxies reflect lower degradation of lignin than lipids, for instance, due to the limited abundance of specialized lignin-degrading microorganisms. Alternatively, the differences might stem from the stability of lignin degradation proxies during decomposition under anoxic conditions[52] or abiotic alteration of proxies over long time frames. Other possibilities include an effect of hydrodynamic sorting. A previous study has observed differences in lignin- and lipid-based degradation proxies between ESAS sediment fractions, with more degraded, lipid-rich organic matter associated with fine-grained sediments, and less

degraded, lignin-rich organic matter in coarser sediments[53]; changes in depositional regime could thus differently affect lignin- and lipid-based degradation proxies.

High-resolution samples from above and below the IBPT represent a continuum from organic matter that has been constantly frozen since the Pleistocene, to organic matter that has thawed in recent decades. Linear interpolation between IBPT positions measured in 1982/3 and 2013 suggests the onset of thaw at the upper limit of the high-resolution sections only about 10 and 20 years before drilling in cores 4D-13 and 2D-13. We examined potential changes in degradation proxies over this time by comparing organic matter properties above and below the IBPT and testing for correlations with distance from the IBPT in the thawed core part. The few significant effects were non-systematic in direction (Supplementary Table 4), and occurred abruptly (Supplementary Fig. 1). This indicates that the observed variability in degradation proxies was likely driven by differences in the degradation state of organic matter at the time of deposition rather than by decomposition upon thaw. However, the data do not exclude organic matter decomposition after thaw per se; changes in degradation proxies could be masked by source variability or might become detectable only after longer periods of thaw and degradation.

### Greenhouse gas production in recently thawed subsea strata

Microbial degradation of organic matter following subsea permafrost thaw might lead to greenhouse gas release. During anoxic incubation at 4 °C, $CO_2$ production from thawed subsea permafrost was highest at the beginning (58–250 μmol g$^{-1}$ OC d$^{-1}$) and decreased over time. The average rate was 2.4 μmol g$^{-1}$ OC d$^{-1}$ (0.7–6.4 μmol g$^{-1}$ OC d$^{-1}$) over the entire period of 600 days (Supplementary Fig. 2 and Supplementary Table 5). Very weak $N_2O$ production was observed after the $CO_2$ peak, in the first 1–5 day interval for samples 4D-13, 2D-13, and 4D-12 and in the 9–15 day interval for 1D-14 (2–55 nmol g$^{-1}$ OC d$^{-1}$). Nitrous oxide production was followed by net $N_2O$ consumption to concentrations below the detection limit for the rest of the incubation (Supplementary

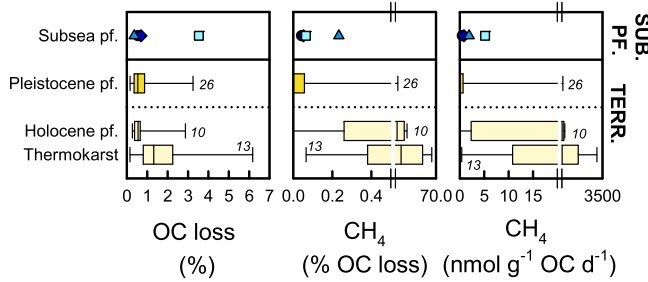

**Fig. 7 | Comparison of greenhouse gas production from subsea permafrost with terrestrial permafrost.** Shown are organic carbon (OC) loss, the contribution of CH$_4$ production to organic carbon loss, and absolute rates of CH$_4$ production during the first year of incubation. Subsea permafrost (sub. pf.) data are from cores 4D-13, 2D-13, 4D-12, and 1D-14 from the Buor-Khaya Bay, and data from previous studies include terrestrial, Pleistocene, and Holocene permafrost (pf.) and well as thermokarst deposits[10,22,25]. For comparability, only samples with organic carbon content <5% and incubation temperature at 4 °C are included. Box plots show medians with 25th and 75th percentiles as box limits, minimum/maximum values as whiskers, and numbers of observations in italics.

Table 5). Methane production was first observed for the 1−5 day interval in the 4D-13 sample, and for the 9−16 day interval in the 2D-13, 4D-13 and 1D-14 samples (4−38 nmol g$^{-1}$ OC d$^{-1}$; Supplementary Fig. 2 and Supplementary Table 5). In each case, the first occurrence of CH$_4$ also represented the peak in CH$_4$ production rates that then decreased to <3 nmol g$^{-1}$ OC d$^{-1}$ after 600 days. On average, CH$_4$ production amounted to 1.7 nmol g$^{-1}$ OC d$^{-1}$ (0.4−4.1 nmol g$^{-1}$ OC d$^{-1}$). The sequence of peaks in CO$_2$, N$_2$O, and CH$_4$, each followed by a decline in net production, points at a transition from electron acceptors of higher to lower redox potential during the incubation.

Both CO$_2$ and CH$_4$ production were well described by two-stage decomposition models. Linear correlation of modeled vs observed cumulative CO$_2$ and CH$_4$ production showed $R^2$ > 0.99 for all samples. This indicates that organic matter degradation dynamics over the incubation period could be well approximated by two discrete components. For CH$_4$, the two model components likely reflected two organic carbon pools of different degradability, and the more easily degradable pool accounted for less than 0.001% of initial organic carbon (see Supplementary Table 6 for fitted parameters). For CO$_2$, the transition to N$_2$O and CH$_4$ production after the CO$_2$ peak suggests not the exhaustion of a more easily degradable carbon pool, but rather a depletion of suitable electron acceptors, behind the two modeled stages of decomposition. Less than 0.2% of the initial organic carbon was mineralized to CO$_2$ in the first decomposition stage (Supplementary Table 6).

Subsea permafrost in the Buor-Khaya Bay showed similar organic carbon losses as terrestrial permafrost, and comparatively low CH$_4$ production. During the first year of incubation, subsea permafrost lost on average 1.3% of organic carbon (range 0.4−3.5%). Previous anoxic, cold-temperature (4 °C) incubations have reported similar losses for terrestrial permafrost of low organic carbon content (<5%), including Pleistocene permafrost, Holocene permafrost, and thermokarst material (Fig. 7)[10,22,25]. During the first year of incubation, CH$_4$ production averaged 0.1% (0.04−0.23%) of organic carbon lost from subsea permafrost, at a rate of 2.1 nmol CH$_4$ g$^{-1}$ OC d$^{-1}$ (0.5−5.2 nmol g$^{-1}$ OC d$^{-1}$). These ranges are comparable to those of Pleistocene permafrost, but lower than those of Holocene permafrost and thermokarst deposits over the same time frame (Fig. 7)[10,22,25]. Previous studies on terrestrial permafrost have however also highlighted high variability in CH$_4$ production rates, frequent CH$_4$ production below the detection limit, and multi-month lag times before its onset[10,22,25,54]. Similarly, high variability in CH$_4$ production has been observed for subglacial deposits of low organic carbon content, with average rates between 0.3 and 1100 nmol CH$_4$ g$^{-1}$ OC d$^{-1}$

during 1 year at 4 °C[55]. We here found the highest CH$_4$ production per organic carbon in samples with a stronger influence of aquatic translocation, indicated by lower organic carbon and nitrogen content, OC/TN and TAR, S/V and C/V lignin phenol ratios suggesting forest sources, as well as higher δ$^{13}$C values and C$_{25}$/(C$_{25}$ + C$_{29}$) $n$-alkane $Sphagnum$ proxies (Spearman's correlation $p$ < 0.1). Potential mechanisms behind variability among subsea permafrost samples and compared to terrestrial permafrost include differences in organic matter degradability, methanogenic microbial community composition and activation after thaw, and pore fluid composition, including electron acceptors of higher redox potential that could inhibit CH$_4$ production or facilitate CH$_4$ consumption.

These constraints on CH$_4$ production by decomposition in thawed subsea permafrost may be compared with estimates of ocean-atmosphere CH$_4$ fluxes. Combining the organic carbon thaw-out rate described above with methanogenesis observed during the experiment results in an estimated production of 3.6 μmol CH$_4$ m$^{-2}$ d$^{-1}$ (standard deviation 4.0 μmol CH$_4$ m$^{-2}$ d$^{-1}$) during the first 600 days after thaw. If methanogenesis persists longer, or even increases over time as observed for terrestrial permafrost[10,22,54], CH$_4$ production by decomposition in thawed subsea permafrost could be substantially larger. The thaw depth of the six cores mentioned in this study, and of two additional cores from the Buor-Khaya Bay[16] averages 14 ± 7 m in 2012−2014. Extrapolating the two-stage model beyond 600 days results in the production of additional 129 μmol CH$_4$ m$^{-2}$ d$^{-1}$ (standard deviation 167 μmol CH$_4$ m$^{-2}$ d$^{-1}$). Since the model converges to constant CH$_4$ production rates once the easily degradable pool is depleted, this extrapolation does not account for the eventual decrease in CH$_4$ production that is expected with the depletion of all carbon pools over longer time frames. While this quantitative assessment establishes a scale for potential CH$_4$ production by decomposition processes in thawed subsea permafrost, the in situ release of CH$_4$ can be influenced by many factors that are not accounted for in our experiment. Methanogenesis rates might be affected by changes of the microbial community over time[56], or by intrusion of seawater, potentially transporting microbial substrates that could facilitate or inhibit CH$_4$ production such as sulfate[28]. A previous meta-analysis reported also a decreasing contribution of CH$_4$ to organic carbon losses with lower temperatures[11]. Methane production might consequently be lower than in our incubation under the in-situ temperatures around zero. On the other hand, part of the produced CH$_4$ is likely aerobically or anaerobically oxidized within thawed subsea permafrost[28,57], in surface sediments and the overlying water, before it reaches the atmosphere. Ocean-atmosphere fluxes of CH$_4$ in the vicinity of the subsea permafrost drilling locations are in the order of 300−1300 μmol m$^{-2}$ d$^{-1}$ based on several years of field observations[26]. The lower rates of CH$_4$ production by subsea permafrost decomposition estimated here, and the likely oxidation of part of this CH$_4$, do not point to a dominant contribution of organic matter decomposition in thawed subsea permafrost to the high emissions observed in the area. We emphasize, however, the high variability of observed CH$_4$ production rates, and the limitations of upscaling from incubations to natural environments. Taken together, the high CH$_4$ emissions ubiquitously observed in the field likely stem from other sources such as pre-formed CH$_4$ in gas pockets in the subsea permafrost, collapsing CH$_4$ hydrates, or venting of a deep thermogenic CH$_4$ pool.

The stable isotopic values of CH$_4$ generated during our incubation fall in line with the range previously reported for microbial fermentation[58]. The δD values averaged −300 ± 14‰, and δ$^{13}$C −65 ± 5‰ (Supplementary Table 7). The here determined fingerprint of CH$_4$ from microbial degradation of subsea permafrost organic matter can be combined with isotopic fingerprints of other potential CH$_4$ sources to calculate the relative contributions of these sources to the CH$_4$ release observed in the field.

**Table 1 | Overview of the subsea permafrost drill cores 4D-13, 2D-13, 4D-12, 1D-14, and 5D-13 from the Buor-Khaya Bay, drilled near Muostakh Island and in Ivashkina Lagoon**

| | 4D-13 | 2D-13 | 4D-12 | 1D-14 | 5D-13 |
|---|---|---|---|---|---|
| Coordinates | 71°37′03″N 129°55′19″E | 71°37′44″N 129°52′53″E | 71°37′46″N 129°52′32″E | 71°45′19″N 129°23′54″ | 71°44′42″N 129°24′17″E |
| Location | 0.6 km from Muostakh | 2.5 km from Muostakh | 2.9 km from Muostakh | Ivashkina Lagoon | Ivashkina Lagoon |
| Inundation time | 145 years | 460 years | 511 years | n.a. | n.a. |
| Water depth | 2.5 m | 3.4 m | 2.5 m | 3.1 m | 3.6 m |
| Core length | 21.0 m | 30.4 m | 55.7 m | 38.2 m | 27.1 m |
| IBPT depth | 8.9 m | 15.9 m | 23.9 m | 12.0 m | 8.2 m |
| IBPT deepening | 18.3 ± 0.1 cm year$^{-1}$ | 9.3 ± 1.7 cm year$^{-1}$ | n.a. | n.a. | n.a. |

Time since inundation, depth of the ice-bonded permafrost table (IBPT) in 2012–2014, and rate of IBPT deepening between 1982/83 and 2013 for 4D-13, 2D-13, and 4D-12 are from ref. [16]. All depths are in m below the sea floor.

In addition to the possible release of the strong greenhouse gas $CH_4$, our incubation experiment suggests that thawing subsea permafrost might be a so-far less considered driver of ocean $CO_2$ emissions and ocean acidification. Using the same approach as for $CH_4$, we estimate the average production of 5.2 mmol $CO_2$ m$^{-2}$ d$^{-1}$ (standard deviation 6.2 mmol $CO_2$ m$^{-2}$ d$^{-1}$) in the first 600 days after thaw, and an additional 201 mmol $CO_2$ m$^{-2}$ d$^{-1}$ (standard deviation 232 mmol $CO_2$ m$^{-2}$ d$^{-1}$) over longer time frames. Previous studies have reported high concentrations of $CO_2$ in the near-coastal Laptev Sea, the net release of $CO_2$ to the atmosphere, and strong ocean acidification by $CO_2$ dissolution threatening ocean fauna, and have linked these to the decomposition of terrigenous organic matter to $CO_2$ in the ocean[32,59]. The respiration of terrigenous organic matter has been estimated to 5.9 mmol C m$^{-2}$ d$^{-1}$ for the outer Laptev Sea[60]. Rates are likely higher in our study area closer to the coast considering the higher concentrations of terrigenous organic matter[32] and water column $CO_2$[59]. Hence, the $CO_2$ flux estimate based on subsea permafrost incubation is of a magnitude relevant for $CO_2$ dynamics in ESAS waters. Under natural conditions, however, inorganic carbon produced by organic matter decomposition in subsea permafrost will be present not only as $CO_2$, but also as carbonate and bicarbonate, in pH-dependent proportions. How much $CO_2$ is eventually released to the water column will also depend on dissolved inorganic carbon consumption by microbial processes and the balance of precipitation and dissolution reactions. Nevertheless, our findings suggest that organic matter decomposition in recently thawed subsea permafrost could also play a role in $CO_2$ emissions and ocean acidification in areas of the rapid thaw of subsea permafrost.

## Synthesis

Subsea permafrost on the extensive Eurasian Arctic Ocean shelf seas is rapidly thawing due to natural and anthropogenic warming. We here characterize organic matter composition and dynamics in a set of subsea permafrost drill cores from the southeast Laptev Sea that reflect sediment deposition in a heterogeneous and dynamic landscape over the past 160,000 ka. Lignin- and lipid-based biomarker proxies indicate the contribution of organic matter originating from tundra and forest to the current thaw front of subsea permafrost near Muostakh Island, and the modification of this organic matter during aquatic transport. Although organic carbon content was here relatively low (average 0.7 ± 0.3%), the high rates of permafrost thaw of 14 ± 3 cm year$^{-1}$ at the study site yield a thaw-out of 1.3 ± 0.6 kg OC m$^{-2}$ year$^{-1}$. These rates exceed those for terrestrial sites by a factor of 35 for the deepening of the permafrost table, and by a factor of nine for organic matter thaw. Constraining the susceptibility of the vast and rapidly thawing subsea permafrost organic carbon pool to degradation is urgently needed for improving estimates of greenhouse gas emissions from all permafrost compartments. This study provides constraints for subsea permafrost on potential $CO_2$, $CH_4$, and $N_2O$ production by organic matter decomposition upon thaw

and the isotopic composition of $CH_4$. Our findings point to other sources than microbial degradation of thawing subsea permafrost as the main drivers of the high $CH_4$ emissions in the study area. Subsea permafrost might however be a contributor to strong ocean acidification in the East Siberian Arctic Shelf region that has not been considered so far.

## Methods

### Study area

The Buor-Khaya Bay is in the southeast Laptev Sea and part of the ESAS, located east of the Lena River Delta (Fig. 1). The area has been affected by marine re- and transgressions during the Pleistocene and Holocene. The inner, most shallow part of the ESAS including the Buor-Khaya Bay was likely exposed to the atmosphere as part of Beringia from at least the onset of the penultimate glaciation ca. 190 ka ago to the end of the Pleistocene[18,49]. Large amounts of organic carbon accumulated in Beringia during the Pleistocene, often in the form of ice-rich, fine-grained permafrost deposits known as Ice Complex deposits (ICD) or Yedoma[3,35] that partly or completely stem from aeolian deposition[35,36], in addition to often coarser-grained permafrost of fluvial and alluvial origin[2]. Pleistocene permafrost deposits were partly degraded by thermokarst formation during warm periods in the MIS 5e interglacial[2] and at the end of the Pleistocene[3], and later by thermal, sea-ice and wave-induced erosion when the ESAS became submerged after the Last Glacial Maximum[3–5]. While Pleistocene permafrost deposits are at least partly preserved on land[2], it is unclear how much of the original permafrost, now underneath the ESAS, still exists[1,3,16,17].

### Fieldwork and drill core description

Subsea permafrost drill cores were obtained during expeditions in spring 2012, 2013, and 2014. Cores were drilled from the sea ice in the Buor-Khaya Bay, using a drilling rig (URB-4T) with a hydraulic rotary-pressure mechanism that operates without drilling fluid to avoid carbon contamination. Well tubes and borehole casings were 4 m long and 147 mm in diameter. Casings were drilled into the seabed, and cores were extracted from the bore holes, sectioned frozen and transported to Tiksi in thermo-insulated boxes for storage at −12 °C. Details on drilling procedure and logistics have been described by Shakhova and co-workers[16].

The five cores used in this study were drilled in the Buor-Khaya Bay (71–72°N, 129–130°E; Fig. 1), specifically on the subsea thermo-erosion terrace of Muostakh Island (4D-13, 2D-13, 4D-12) and in Ivashkina Lagoon (1D-14, 5D-13). Water depths were between 2.5 and 3.6 m at the time of sampling, following submersion 145–511 years ago at this near-coastal location[16] (Table 1). All cores crossed through the IBPT, i.e., the upper part was thawed and the lower part frozen at the time of sampling. Information on core length, IBPT depth at the time of sampling, and IBPT deepening rates as published by Shakhova et al.[16] is available in Table 1. We here present (i) an OSL-derived chronology for

core 4D-12 together with (ii) previously published lithological data[16,33], (iii) organic carbon concentration and isotopic composition at five depths of each core, followed by (iv) a detailed analysis of the organic matter at the current upper thaw front of 4D-13, 2D-13 and 4D-12 at high-resolution, and (iv) an incubation experiment of selected samples to quantify potential greenhouse gas production rates.

## Grain size description

Grain size data for 4D-13, 2D-13, 4D-12, and 1D-14 have been previously published[16,33] and are here discussed again together with data for 5D-13. These data complement the new OSL data to place the subsea permafrost drill cores in the stratigraphic context of coastal permafrost in the study region (see the original publications for method details[16,33]). We here calculated statistical properties of grain size distributions, in particular the number of modes and their peak grain size, using the Gradistat v8 program[61]. To derive the modes, the program first normalizes the fraction of material in each analyzed size class by the difference between the base 2-logarithms of the upper and lower size class thresholds. A mode is then defined as a local peak in the normalized fraction within a size class that reaches at least 15% of the overall largest peak of the respective sample. Original data and results of mode analysis are provided in Supplementary Table 8.

## Optically stimulated luminescence chronology

Subsamples of core 4D-12 were taken under dark room conditions for OSL dating and processed and analyzed at the Lund Luminescence Laboratory, Sweden. A detailed method description is provided in the Supplementary Material.

## Specific surface area of minerals

For analysis of the specific surface area, samples were thawed at room temperature and manually homogenized. Aliquots were freeze-dried, combusted for 12 h at 400 °C to remove organic matter, rinsed with Milli Q water, again freeze-dried and degassed under $N_2$ flow for 2 h at 200° using a Micromeritics FlowPrep 060 Sample Degas System. Specific surface area of minerals was determined with a Micromeritics Gemini VII Surface Area and Porosity analyzer with $N_2$ as absorbent.

## Bulk organic matter properties

For analysis of total organic carbon and total nitrogen concentrations as well as $\delta^{13}C$ values of organic carbon, freeze-dried subsamples were ground in a mortar, and aliquots were filled into Ag capsules, acidified with 1 M HCl, and dried to remove carbonates. The acidification procedure was repeated until effervescence stopped. Samples were analyzed using a Finnigan Delta Plus XP mass spectrometer coupled to a Thermo Fisher Scientific Flash 2000 Isotope Ratio Mass Spectrometer (IRMS) Element Analyzer via a Conflo II interface. Analytical uncertainty was determined for a subset of samples in triplicates (21 of 153 samples). Standard deviations for the triplicates averaged 0.039% of sample dry weight for total organic carbon, 0.003% of sample dry weight for total nitrogen, and 0.158‰ for $\delta^{13}C$. Values of individual samples are presented in Supplementary Table 2.

## Lignin biomarker analysis

Micro-wave assisted CuO oxidation of freeze-dried and ground samples was used to hydrolyze the macromolecules that constitute the bulk of organic matter, and analyze the derived lignin phenols, hydroxybenzenes, and $p$-hydroxybenzenes;[62] see the Supplementary Material for details. Vanillin (Vl), acetovanillone (Vn), vanillic acid (Vd), syringaldehyde (Sl), acetosyringone (Sn), syringic acid (Sd), $p$-coumaric acid ($p$Cd), ferulic acid (Fd), benzoic acid (Bd), $m$-hydroxybenzoic acid ($m$-Bd), 3,5-dihydroxybenzoic acid (3,5-Bd), $p$-hydroxybenzaldehyde (Pl), $p$-hydroxyacetophenone (Pn), and $p$-hydroxybenzoic acid (Pd) were quantified. The sum of vanillyl

phenols (V) was calculated as Vl + Vn + Vd, the sum of syringyl phenols (S) as Sl + Sn + Sd, the sum of cinnamyl phenols (C) as $p$Cd + Fd, and the sum of all lignin phenols as V + S + C. Individual values are presented in Supplementary Table 9.

Five ratios were calculated to describe the origin and decomposition state of lignin. Ratios of syringyl over vanillyl subunits (S/V) and cinnamyl over vanillyl subunits (C/V) reflect the source of lignin phenols, with higher S/V ratios in angiosperm than in gymnosperm tissues, and higher C/V ratios in non-woody (e.g., leaves) than in woody tissues[63]. Ratios of acids over aldehydes of syringyl and vanillyl subunits (Sd/Sl, Vd/Vl) reflect the decomposition state of lignin; both increase during aerobic decomposition due to oxidation[64]. Similarly, ratios of 3,5-dihydroxybenzoic acid over vanillyl (3,5-Bd/V) increase during decomposition[65].

## Lipid biomarker analysis

Lipid biomarkers were extracted from freeze dried samples with accelerated solvent extraction and analyzed for $n$-alkanes, $n$-alkanoic acids, $n$-alkanols, and steroids using gas chromatography-mass spectrometry (see Supplementary Material for details). Individual values are presented in Supplementary Tables 10–12.

We here present concentrations of HMW $n$-alkanes ($C_{25}$-$C_{33}$), HMW $n$-alkanoic acids ($C_{24}$-$C_{30}$), and HMW $n$-alkanols ($C_{24}$-$C_{32}$), together with a suite of ratios to describe the origin and degradation state of lipids. The terrigenous-aquatic ratio (TAR) was calculated as the ratio of the $n$-alkanes ($C_{27} + C_{29} + C_{31}$)/($C_{15} + C_{17} + C_{19}$) and indicates terrestrial vs aquatic origin of lipids based on the high abundance of $C_{27}$, $C_{29}$, and $C_{31}$ in land-plant epicuticular waxes vs the high abundance of $C_{17}$ in algae[41]. The $n$-alkane ratio $C_{25}/(C_{25} + C_{29})$ was used as an indicator of lipids derived from *Sphagnum* mosses[66]. The carbon preference indices (CPIs) of HMW $n$-alkanes ($C_{25}$–$C_{31}$) and $n$-alkanoic acids ($C_{24}$–$C_{28}$) decrease during degradation approaching one[67]. Ratios of HMW $n$-alkanoic acids ($C_{24}$–$C_{28}$) over $n$-alkanes ($C_{25}$–$C_{31}$), and of HMW $n$-alkanols ($C_{24}$–$C_{32}$) over $n$-alkanes ($C_{25}$–$C_{31}$) decrease during decomposition due to the higher stability of $n$-alkanes, and ratios of sitostanol over $\beta$-sitosterol increase due to conversion of stenols to stanols[67].

## Incubation experiment

The incubation experiment was designed to measure production rates of $CH_4$, $CO_2$, and $N_2O$ by decomposition of thawed subsea permafrost organic matter under conditions as close as possible to the natural environment (i.e., cold and anoxic), and to determine the $^{13}C$- and D-isotopic composition of produced $CH_4$. One sample each from four cores was chosen randomly for the experiment, covering a range of depths: 4D-13 at 4.55 m depth, 4D-12 at 27.4 m, 2D-13 at 24.7 m, and 1D-14 at 25.1 m. The subsea permafrost material was incubated at 4 °C under anoxic conditions and concentrations of greenhouse gases were determined on days 1, 5, 9, 16, 23, 37, 114, 286, and 601. The isotopic composition of accumulated $CH_4$ was analyzed after 335 days of incubation at the Institute for Marine and Atmospheric Research at Utrecht University. A detailed method description, including of model fitting to observed greenhouse gas production, is provided in the Supplementary Material. Data on greenhouse gas production rates, model fits, and isotopic composition are presented in Supplementary Tables 5–7.

## Statistical analyses

Statistical analyses were performed using R 3.3.1[68]. We tested correlations between measured parameters across the three cores using Spearman's rank sum correlations. This method tests for a monotonous relationship between two parameters; the closeness of this relationship is then described by Spearman's correlation coefficient $\rho$. Differences between core segments above and below the IBPT were tested individually for each core using non-parametric Wilcox tests. Spearman's rank sum correlations were further applied to test for

monotonous changes with distance from the IBPT in the thawed part of the core that might reflect progressing decomposition upon thaw. Statistical tests were considered significant at $p < 0.05$.

## Data availability

All data generated for this publication are presented in detail in the Supplementary Information. The data on carbon and nitrogen concentration and isotopic composition, as well as biomarker concentrations, have been additionally deposited in the Stockholm University Bolin Centre Database (https://bolin.su.se/data/wild-2022-subsea-permafrost-1).

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

## Acknowledgements

This study was funded by the Swedish Research Council VR (grant numbers 621-2013-5297 and 2017-01601 to Ö.G., as well as 2018-05489 to B.W.), the Swedish Research Council for Sustainable Development Formas (grant number 2018-01547 to B.W.), the European Research Council (ERC-AdG CC-TOP, grant number 695331 to Ö.G.), and the Russian Science Foundation (grant number 21-77-30001 to I.S.). Fieldwork was supported by the Russian Ministry of Science and Higher Education (project 075-15-2020-928 to N.S., and theme 0211-2021-0010 to O.D.). This study was further supported by the Tomsk State University Development Programme (Priority-2030). We further thank Rajendra Shrestha (Lund University) for OSL sample preparation, and Carina van

der Veen (Institute for Marine and Atmospheric Research Utrecht) for isotopic analysis of $CH_4$.

## Author contributions

B.W., I.S., and Ö.G. designed the study; O.D., A.R., D.K., V.T. as well as I.S. conducted the drilling campaigns and processed cores and samples. H.A. and M.J. provided OSL data. Analysis of bulk organic matter properties was performed by T.T., of lignin phenols by B.W., of lipids by I.N. and F.M., and of specific surface area by H.G. B.W. and Ö.G. led the writing of the manuscript with the contribution of all authors N.S., O.D., A.R., D.K., V.T., T.T., H.J., I.N., F.M., H.A., M.J., A.M., and I.S.

## Funding

## Competing interests

The authors declare no competing interests.
