## [Peer Review File · Nature Communications]

Organic matter composition and greenhouse gas production of thawing subsea permafrost in the Laptev SeaReviewers' Comments:

Reviewer #1:

Remarks to the Author:

Wild et al. sought to improve our understanding of subsea permafrost organic matter pools, and the potential of subsea permafrost to becoming a source of greenhouse gas to the atmosphere after thaw. This work is valuable because subsea permafrost is more difficult to access than terrestrial permafrost, making observational data scarce and basic properties unknown or poorly constrained.

Wild et al. found that the organic matter in subsea permafrost cores from the Laptev Sea originate from forest, tundra, and peatlands, and were deposited over 160,000 years ago. They also report that the annual rate of thaw is nine-fold higher in subsea permafrost relative to thaw rates in terrestrial permafrost. These results indicate that subsea permafrost stores old carbon that is now potentially vulnerable to microbial decomposition since thaw is occurring at a fast pace. These are noteworthy results that have important significance to global models that aim to identify and quantify carbon feedbacks to climate change.

Though the findings reported are very compelling and a great contribution to the field, the abstract and introduction need to be re-written to clearly identifying goals and hypotheses, as well as to set the stage for the implications of the reported results.

For example:

- a) the objectives of this study are not clearly stated and are often vague, and there is a weak link between the objectives and findings (see detailed comments in the abstract section below),
- b) there were no clear hypotheses stated in the introduction, even though they are alluded to in the results/discussion.
- c) the implication of the age and origin of subsea permafrost deposits to carbon dynamics and climate-feedbacks is not discussed in the introduction, which makes it difficult for the reader to understand the context and importance of these findings.

There was a large focus on CH₄ sources and emissions in the introduction, but little on CO₂ and N₂O. Since the results do not suggest large CH₄ emissions, the introduction would benefit from a discussion on different greenhouse gas emission pathways, instead.

Overall, the writing should be more polished to avoid run-on sentences that make it difficult for the reader to understand the authors' message.

Abstract

The objectives of the paper are not clearly stated and are often vague. There is a weak link between the objectives and findings. I had to read the abstract and introduction a few times. Here is an excerpt from the abstract to highlight this issue:

i) Objective: Identify source of organic matter

a. Result: Grain sizes, optically-stimulated luminescence and biomarkers suggest deposition of aeolian silt and fluvial sand over 160 000 years, with dominant fluvial/alluvial deposition of organic matter originating in forests, tundra and peatlands.

This is fine.

ii) Objective: Identify degradation state of organic matter

a. Result: We estimate the annual thaw-out of 1.3 ± 0.6 kg organic carbon m⁻² in subsea permafrost in the study area, nine-fold exceeding rates for terrestrial permafrost.

Here, the result is not addressing the degradation state of organic matter (which to me indicates vulnerability of organic matter to decomposition), but rather it indicates the degradation state of

permafrost.

iii) Objective: Potential greenhouse gas production upon thaw

a. Result: Based on 10- 34 month incubations, we constrain potential CH₄, CO₂ and N₂O production by organic matter decomposition after thaw

I'm not sure what "constrain" means. Did the authors mean quantify gas emissions? Why not report the magnitude of C loss in the results portion of the abstract?

"...which combined with the CH₄ isotopic fingerprint, facilitates the apportionment of the high CH₄ fluxes and strong ocean acidification in the region between subsea permafrost and other sources."

This seems like gas emissions were partitioned by sources. Please state what these sources are.

Finally, It would be helpful for the authors to report results on the magnitude of greenhouse gas emissions in the abstract to support the claims and conclusions about subsea permafrost and its potential to become part of the permafrost-carbon-climate feedback.

Specific comments:

Line 26: Please add comma before "or become." This sentence will be a little clearer if edited to: "...pool that might be, or will become, a significant..."

Lines 28-29: It seems like source of organic matter is a separate topic that shouldn't be combined with degradation state. These objectives were listed separately in the introduction—I think it's more helpful to have the 3 objectives...or be consistent.

Line 32: Change to annual thaw rate rather than thaw-out.

Lines 33-37: This sentence is not very clear. Instead of "constraining [gas production] by organic matter decomposition after thaw," please clarify. Is the goal to constrain the source of gas production, or is the goal to constrain the source of fluvial/alluvial deposition of organic matter? Also, please clarify what the sources are.

Introduction

In the introduction, it would be nice if the authors gave the reader some more context about subsea permafrost since it is not studied as much as terrestrial permafrost. Relative to terrestrial C pools, how much C is vulnerable to decomposition? Also, what are the loss pathways and how does ocean water affect how much C is actually emitted to the atmosphere?

The objectives stated in the introduction do not match the objectives in the abstract. Also, as it is written, it is difficult to keep track of which method addressed which objective. It would help if the authors identified in the sentences that describe sample collection and methods (Lines 87-97) how these addressed each objective.

One of the main objectives of the paper is to describe the source of subsea permafrost deposits. However, it isn't until reading the results/discussion that the authors identify the potential sources. This needs to occur in the introduction, again with some context as to why the source matters.

Specific comments:

Line 50: The word "remobilizing" is not very clear: does it mean old C is lost due to higher microbial activity after thaw, or is this more of a physical disturbance due to erosion?

Lines 51-53: Please address this run-on sentence, and add citations.

Lines 65-66: How much faster is this rate of thaw relative to terrestrial permafrost?

Lines 87-88: "Fast ice" was not defined—is this ice in the surface layers, or is it ice that is currently melting at a faster rate?

Results/Discussion

Lines 120-121: I had to re-read the introduction to confirm that there were no hypotheses stated. I'm unsure how these results are linked to the hypothesis that the subsea permafrost of this site does not represent Ice Complex deposits. There is also very little text in the introduction on the implication of the source and age of deposits to carbon dynamics and climate-feedbacks.

Lines 146-148: What is the estimated magnitude of this organic matter pool?

Conclusion

Line 357-360: The conclusion on greenhouse gas emission is too vague based on the details provided in the results/discussion section.

Reviewer #2:

Remarks to the Author:

Subsea permafrost hosts large, yet poorly quantified amounts of organic carbon (OC) in Arctic sediments. The global warming-induced thawing of these permafrost stores makes its OC vulnerable to microbial decay, potentially producing large quantities of CH₄ with important implications for Arctic carbon cycling and climate. However, due to the scarcity of direct observational data, major uncertainties exist regarding the amount, reactivity, and specific degradation pathways of the thawing subsea permafrost. These critical knowledge gaps seriously compromise our ability to assess the current and future contribution of permafrost-derived CH₄ to the global greenhouse gas budget and its feedback on climate.

Wild et al., present unique and thus highly original observations from subsea permafrost drilled on the Laptev Sea shelf. They combine isotopic and biomarker analysis with 10-months incubation experiments to investigate the sources and degradation state of permafrost organic matter and its susceptibility towards microbial decay. They thus directly address one of these critical knowledge gaps and provide novel and important quantitative information that is desperately needed to evaluate the impact of current and projected subsea permafrost thaw on global climate evolution.

Therefore, the manuscript makes an important step towards a better quantification of subsea permafrost-climate feedbacks. It provides noteworthy results that are of significance not only to the specific field, but also to the wider carbon cycle/climate community. The methodology looks overall sound and is described in sufficient detail. Please find below detailed comments concerning data analysis, interpretation, and conclusions:

Environmental settings:

p.7, l. 171: How does the reported permafrost thawing rate of 14 cm/yr compare to other estimates of subsea permafrost thaw on the Siberian Shelf?

How representative are local environmental conditions for permafrost-bearing sediments on the Arctic shelf? In other words, how transferable might results be?

Sources of OM

L.160-162. Bulk OC contents are low and authors thus assume that OM is derived from fluvial/alluvial sediment deposits. This seems to be confirmed by $\delta^{13}\text{C}$ values (L.186-187), which range between -25.6 – -24.3 ‰. However, one could argue that the observed $\delta^{13}\text{C}$ range still falls within the range of terrestrial OM(?!). Could one not assume that the difference between Ice Complex Deposits and Thermokarsts ($\delta^{13}\text{C} < -26\text{‰}$) and the observed subsea permafrost samples be caused by alteration during transport (eg via rivers) and not necessarily distinct sources?

Then in L.205-207, the authors conclude that the TAR of 32 ± 16 indicates the dominance of OM from lower plants such as algae and mosses, thus supporting the statement on L.193-196. However, this seems to be a bit of an oversimplified interpretation of (n-alkanes) TAR = $\text{C}_{27} + \text{C}_{29} + \text{C}_{31} / \text{C}_{15} + \text{C}_{17} + \text{C}_{19}$, HMW homologs from (higher) plant waxes vs. LMW from aquatic sources. Looking at Table S9, most of C_{27} - C_{29} - C_{31} values greatly outweigh those of LMW, thus contradicting an aquatic predominance.

Then the ratio $\text{C}_{25} / \text{C}_{25} + \text{C}_{29}$ is used to show the relative contributions of Sphagnum mosses (peat-forming) to OC pool (L.209 onwards). The low ratio values do support the low contribution of Sphagnum relative to leaf waxes (C_{29}), but again I do not really see how this would indicate the predominance of aquatic sources (L.210).

In summary, the interpretation of n-alkane TAR would benefit from some clarification and a more inclusive discussion of what the actual values may indicate. Lipids represent a relatively minor pool of bulk OM. The TAR values presented here support a shift from TAR values found in ICD and thermokarst to those determined in the subsea permafrost, which may reflect modified during transport (is that what you mean by aquatic translocation?), rather than a predominance of aquatic sources.

Greenhouse gases production

Is there any link between CH_4 (and CO_2) production and OM composition in these incubations? Could that explain the differences in rates and timings of initial production (L.284-286)?

Could the sediment and/or pore fluid compositions offer further explanations for the comparably low CH_4 production rates and the differences in CO_2/CH_4 production (L.289)? Upon sediment thaw during incubation, some of these compositional features (e.g., background sulfate, nitrate, or iron hydroxides) could inhibit CH_4 production or consume produced CH_4 ?

Also, the result that subsea permafrost CH_4 production rates are lower than in terrestrial permafrost incubations is an important and novel finding. Would it be possible to identify potential reasons? For instance, what role do thermodynamic limitations (other than temperature), microbial biomass dynamics (also activation from the resting stage), microbial community composition, etc play? Would you expect results to be transferable to other subsea permafrost settings?

Wider Impact

The authors should discuss the potential wider impact of their findings in more detail and more comprehensively.

For instance, I would like to strongly encourage the authors to provide a quantitative measure (e.g. first-order rate constant/turn-over time, multi-pool model, reactive continuum) of the subsea permafrost reactivity. This would not only allow a straightforward comparison of the results with the wide range of published OC reactivity data from different settings, but could also inform the set-up and parametrization of regional models. In addition, it would be interesting to compare the observed methanogenesis rates with observations from other extreme environments (e.g. subglacial environments, the deep biosphere).

If permafrost OC degradation rates are low, seawater sulphate, diffusing downcore from the sediment-

water interface might penetrate into the thawed layer and inhibit methanogenesis, thus limiting CH₄ production.

The authors also compare measured CH₄ production rates with ocean CH₄ emissions. However, a plethora of past studies have shown that, generally, strong CH₄ sinks in both ocean and sediment efficiently consume a large fraction of deep CH₄ fluxes. Thus, one would expect production rates to be higher than observed CH₄ emissions at the ocean surface.

The influence of thawing subsea permafrost on the ocean's carbonate system and saturation state is somewhat (over?)simplified. It will depend on the complex and dynamic interplay of different early diagenetic processes. While aerobic degradation of OC (for instance in the water column or the shallow sediment layers) would indeed result in the acidification of overlaying waters, the anaerobic degradation of OC in deeper sediment layers and the multitude of strongly coupled redox processes make it difficult to predict the possible impact that will depend on the delicate balance between aerobic and (an)aerobic CH₄ oxidation processes, as well as a number of mineral precipitation/dissolution processes.

Organic matter composition and greenhouse gas production of thawing subsea permafrost in the Laptev Sea

Birgit Wild, Natalia Shakhova, Oleg Dudarev, Alexey Ruban, Denis Kosmach, Vladimir Tumskey, Tommaso Tesi, Hanna Joss, Inna Nybom, Felipe Matsubara, Helena Alexanderson, Martin Jakobsson, Alexey Mazurov, Igor Semiletov, Örjan Gustafsson

We thank both reviewers for the positive feedback and constructive suggestions. We naturally agree with the reviewers on the “*critical knowledge gaps*” around subsea permafrost with repercussions for global greenhouse gas fluxes and that “*quantitative information ... is desperately needed to evaluate the impact of current and projected subsea permafrost thaw on global climate evolution*”. We address all reviewer comments below in detail, but want to especially highlight two major changes in response to review comments:

- **The objectives and hypotheses have been revised.** Suggested parts of the Results and Discussion section have also been restructured to more clearly connect to the study objectives.
- We have developed and added **two-pool decomposition models** to describe the production of CO₂ and CH₄ during our incubation, to facilitate comparisons and model integration. This includes additional greenhouse gas measurements that now **extend the incubation experiment from 10 to 20 months**. While there are no fundamental changes to interpretations, we substantially refined the discussion of greenhouse gas emissions accordingly, following also other suggestions of the reviewers on that topic.

All reviewer comments are presented below in blue italics, with author responses in black normal font.

Reviewer #1 Comments

Wild et al. sought to improve our understanding of subsea permafrost organic matter pools, and the potential of subsea permafrost to becoming a source of greenhouse gas to the atmosphere after thaw. This work is valuable because subsea permafrost is more difficult to access than terrestrial permafrost, making observational data scarce and basic properties unknown or poorly constrained.

Response: We appreciate the recognition of the unique value of this study, in delivering quantitative data on organic matter composition and vulnerability in the extremely elusive yet potentially important subsea permafrost compartment. We also appreciate all the other positive feedback and constructive suggestions. All comments are addressed in detail below.

Wild et al. found that the organic matter in subsea permafrost cores from the Laptev Sea originate from forest, tundra, and peatlands, and were deposited over 160,000 years ago. They also report that the annual rate of thaw is nine-fold higher in subsea permafrost relative to thaw rates in terrestrial permafrost. These results indicate that subsea permafrost stores old carbon that is now potentially vulnerable to microbial decomposition since thaw is occurring at a fast pace. These are noteworthy results that have important significance to global models that aim to identify and quantify carbon feedbacks to climate change.

Though the findings reported are very compelling and a great contribution to the field, the abstract and introduction need to be re-written to clearly identifying goals and hypotheses, as well as to set the stage for the implications of the reported results.

Response: We agree with the reviewer and adjusted the Abstract and Introduction accordingly – see here briefly on the examples given by the reviewer and in detail on all raised points below.

For example:

a) the objectives of this study are not clearly stated and are often vague, and there is a weak link between the objectives and findings (see detailed comments in the abstract section below),

Response: We clarified our objectives in the Introduction and briefly describe the corresponding approach immediately after each (lines 96-116). Each objective is now aligned with 1-2 sub-chapters of the Results and Discussion section. See also our detailed response to the corresponding reviewer comment below.

b) there were no clear hypotheses stated in the introduction, even though they are alluded to in the results/discussion.

Response: We added hypotheses to Objectives 1 and 3 (lines 97, 111-112) that are then addressed in the Results and Discussion. For Objectives 2 and 4, we specify that the goal was to provide quantitative data (lines 108-109, 113-114).

c) the implication of the age and origin of subsea permafrost deposits to carbon dynamics and climate-feedbacks is not discussed in the introduction, which makes it difficult for the reader to understand the context and importance of these findings.

Response: We added our motivation for targeting organic matter sources and deposition history of subsea permafrost to the Introduction (lines 97-102).

There was a large focus on CH₄ sources and emissions in the introduction, but little on CO₂ and N₂O. Since the results do not suggest large CH₄ emissions, the introduction would benefit from a discussion on different greenhouse gas emission pathways, instead.

Response: The main motivation for our incubation experiment were the high CH₄ concentrations previously observed in ESAS bottom waters and the question to what extent organic matter decomposition in thawed subsea permafrost might contribute to them. The rather low CH₄ production rates in subsea permafrost consequently represent a major finding of this study, and point to other sources behind the high CH₄ concentrations observed in ESAS waters. We therefore decided to keep the section on CH₄ dynamics on the ESAS in the Introduction. However, we agree, and have therefore widened the Introduction to include also CO₂ production during anoxic decomposition (lines 74-78) and introduce the previously observed elevated CO₂ concentrations and ocean acidification on the ESAS (lines 85-91).

Overall, the writing should be more polished to avoid run-on sentences that make it difficult for the reader to understand the authors' message.

Response: We scrutinized the entire manuscript and split sentences to avoid run-ons.

Abstract

The objectives of the paper are not clearly stated and are often vague. There is a weak link between the objectives and findings. I had to read the abstract and introduction a few times. Here is an excerpt from the abstract to highlight this issue:

i) Objective: Identify source of organic matter

a. Result: Grain sizes, optically-stimulated luminescence and biomarkers suggest deposition of aeolian silt and fluvial sand over 160 000 years, with dominant fluvial/alluvial deposition of organic matter originating in forests, tundra and peatlands.

This is fine.

ii) Objective: Identify degradation state of organic matter

a. Result: We estimate the annual thaw-out of 1.3 ± 0.6 kg organic carbon m⁻² in subsea permafrost in the study area, nine-fold exceeding rates for terrestrial permafrost.

Here, the result is not addressing the degradation state of organic matter (which to me indicates

vulnerability of organic matter to decomposition), but rather it indicates the degradation state of permafrost.

iii) Objective: Potential greenhouse gas production upon thaw

a. Result: Based on 10-month incubations, we constrain potential CH₄, CO₂ and N₂O production by organic matter decomposition after thaw

I'm not sure what "constrain" means. Did the authors mean quantify gas emissions? Why not report the magnitude of C loss in the results portion of the abstract?

Response: As addressed also above, we agree that the objectives and their connection to main findings were insufficiently clear in the previous version of the manuscript. We now revised the objectives, defined clear hypotheses and identified where the goal was to provide quantitative data. We also re-structured the Results and Discussion section to clarify the connection to the objectives. We briefly present the updated objectives below, and in detail in lines 96-116.

Objective 1: Constrain the source of organic matter at the current thaw front, and test the hypothesis that subsea permafrost does not represent ICD but deeper deposits. Lithological data, optically stimulated luminescence dating and biomarkers support this hypothesis and suggest dominant fluvial/alluvial deposition of organic matter originating in forests and tundra. Findings are discussed in the Results and Discussion sections "*Stratigraphic context*" (lines 119-149) and "*Organic matter sources*" (lines 151-224).

Objective 2: Quantify organic carbon storage and thaw to provide quantitative constraints for modelling efforts. We show the average annual thaw of 1.3 ± 0.6 kg OC m⁻² in the study area; this rate exceeds that of land-based permafrost in north-east Siberia 9-fold (discussed in "*Organic carbon storage and thaw*"; lines 226-238)

Objective 3: Assess the degradation state of organic matter at the current thaw front, hypothesizing that recommencing organic matter degradation after thaw would be reflected in changing molecular properties of lignin and lipid compound classes. This hypothesis was not supported; high variability in organic matter degradation state along cores was rather related to differences in decomposition before deposition. Changes in degradation proxies after subsea permafrost thaw might be masked by high source variability or become detectable after longer periods of thaw (discussed in "*Degradation state of organic matter at the thaw-front*"; lines 240-274).

Objective 4: Quantify potential rates of CH₄, CO₂ and N₂O production by organic matter decomposition after thaw, in a 20-month incubation experiment. Production of CH₄ and CO₂ averaged 1.7 nmol and 2.4 μmol g⁻¹ OC d⁻¹ whereas that of N₂O was negligible. We discuss these rates in comparison with those from terrestrial permafrost, and with CH₄ fluxes and ocean acidification in the study region (discussed in "*Greenhouse gas production in recently thawed subsea strata*"; lines 276-368).

"...which combined with the CH₄ isotopic fingerprint, facilitates the apportionment of the high CH₄ fluxes and strong ocean acidification in the region between subsea permafrost and other sources." This seems like gas emissions were partitioned by sources. Please state what these sources are.

Response: We removed the reference to CH₄ sources here to avoid confusion and remain within the word limit of the abstract. Potential CH₄ sources in the study region are instead described in the Introduction (lines 81-85) and again addressed in the Results and Discussion section (lines 344-346).

Finally, It would be helpful for the authors to report results on the magnitude of greenhouse gas emissions in the abstract to support the claims and conclusions about subsea permafrost and its potential to become part of the permafrost-carbon-climate feedback.

Response: We added average rates of CO₂ and CH₄ production in the abstract; N₂O was removed from the abstract since rates were extremely low (lines 32-33).

Specific comments:

Line 26: Please at comma before "or become." This sentence will be a little clearer if edited to: "...pool that might be, or will become, a significant..."

Response: Changed as suggested (line 25).

Lines 28-29: It seems like source of organic matter is a separate topic that shouldn't be combined with degradation state. These objectives were listed separately in the introduction—I think it's more helpful to have the 3 objectives...or be consistent.

Response: We now address all four updated objectives in the abstract, but present them in aggregated form to remain within the tight word limit of the abstract (lines 26-28).

Line 32: Change to annual thaw rate rather than thaw-out.

Response: Changed as suggested (line 31).

Lines 33-37: This sentence is not very clear. Instead of “constraining [gas production] by organic matter decomposition after thaw,” please clarify. Is the goal to constrain the source of gas production, or is the goal to constrain the source of fluvial/alluvial deposition of organic matter? Also, please clarify what the sources are.

Response: We changed this sentence to: “During 20-month incubations, CH_4 and CO_2 production averaged 1.7 nmol and $2.4 \mu\text{mol g}^{-1} \text{ OC d}^{-1}$, providing a baseline to assess the contribution of subsea permafrost to the high CH_4 fluxes and strong ocean acidification observed in the region.” (lines 32-34). We removed the reference to other CH_4 sources to avoid confusion and stay within the word limit of the abstract, but describe them in the Introduction.

Introduction

In the introduction, it would be nice if the authors gave the reader some more context about subsea permafrost since it is not studied as much as terrestrial permafrost. Relative to terrestrial C pools, how much C is vulnerable to decomposition? Also, what are the loss pathways and how does ocean water affect how much C is actually emitted to the atmosphere?

Response: These are all excellent questions, and we as a community do not really have robust answers to any of them. Observational data on subsea permafrost are extremely scarce and there is not even an observation-based estimate for the current subsea permafrost organic carbon pool. Based largely on Soviet/Russian investigations and geological system assessments, it is likely that there is substantial subsea permafrost and methane sources in this vast region. A key motivation of this study is to provide observational data to contribute towards answering these excellent questions by the reviewer. We now highlight and elaborate some more on these unknowns in lines 59-62.

The objectives stated in the introduction do not match the objectives in the abstract. Also, as it is written, it is difficult to keep track of which method addressed which objective. It would help if the authors identified in the sentences that describe sample collection and methods (Lines 87-97) how these addressed each objective.

Response: We clarified the objectives (see above for details on the new alignment), re-structured their description to present associated methods immediately after each objective (lines 96-116), and aligned the abstract with the updated objectives, yet in aggregated form to remain within the word limit (lines 26-28).

One of the main objectives of the paper is to describe the source of subsea permafrost deposits. However, it isn't until reading the results/discussion that the authors identify the potential sources. This needs to occur in the introduction, again with some context as to why the source matters.

Response: We now added a description of different Pleistocene permafrost deposits to the Introduction (lines 44-48) and clarify our motivation for targeting subsea permafrost sources as one of the objectives for this study (lines 97-102).

Specific comments:

Line 50: The word “remobilizing” is not very clear: does it mean old C is lost due to higher microbial activity after thaw, or is this more of a physical disturbance due to erosion?

Response: It is the latter. To clarify, we re-phrased to: “... leading to the inundation, transfer, and potentially mineralization, of substantial amounts of old carbon.” (lines 51-53)

Lines 51-53: Please address this run-on sentence, and add citations.

Response: We split the sentence as suggested and added references (lines 54-57).

Lines 65-66: How much faster is this rate of thaw relative to terrestrial permafrost?

Response: It is about 35 times faster. We added this comparison with terrestrial permafrost in central and eastern Siberian (average thaw 0.4 cm year⁻¹ 1990-2020)^{1,2} to the Introduction (lines 70-71); see also the Discussion on organic carbon thaw in subsea vs terrestrial permafrost (lines 233-238).

Lines 87-88: “Fast ice” was not defined—is this ice in the surface layers, or is it ice that is currently melting at a faster rate?

Response: “Fast ice” defines sea ice that is attached to the coastline or sea floor. We changed to the more general and wider term “sea ice” to avoid confusion (lines 95, 407).

Results/Discussion

Lines 120-121: I had to re-read the introduction to confirm that there were no hypotheses stated. I’m unsure how these results are linked to the hypothesis that the subsea permafrost of this site does not represent Ice Complex deposits. There is also very little text in the introduction on the implication of the source and age of deposits to carbon dynamics and climate-feedbacks.

Response: See above; we extended the Introduction of Ice Complex and other Pleistocene permafrost deposits (lines 44-48), explain our motivation for addressing subsea permafrost sources as one of our study objectives (lines 97-102-103), and specify the hypothesis that subsea permafrost does not represent Ice Complex but older deposits (lines 96-97).

Lines 146-148: What is the estimated magnitude of this organic matter pool?

Response: See above – we are not aware of an observation- or model-based estimate of the subsea permafrost organic carbon pool in the peer-reviewed literature. However, a pool size of several hundred Pg organic carbon seems likely, given the expected extent of subsea permafrost. A central motivation of this study is to contribute observational data to facilitate a first solid estimate of the subsea permafrost organic carbon stock. We now clarify the current lack of an estimate in lines 59-62.

Conclusion

Line 357-360: The conclusion on greenhouse gas emission is too vague based on the details provided in the results/discussion section.

Response: We agree and added more detail on the implications of our study for estimating greenhouse gas emissions from subsea permafrost: “Our findings point at other sources than microbial degradation of thawing subsea permafrost as main drivers to the high CH₄ emissions in the study area. Subsea permafrost might however be a contributor to strong ocean acidification in the East Siberian Arctic Shelf region that has not been considered so far.” (lines 384-387)

Reviewer #2 Comments

Subsea permafrost hosts large, yet poorly quantified amounts of organic carbon (OC) in Arctic sediments. The global warming-induced thawing of these permafrost stores makes its OC vulnerable to microbial decay, potentially producing large quantities of CH₄ with important implications for Arctic carbon cycling and climate. However, due to the scarcity of direct observational data, major uncertainties exists regarding the amount, reactivity, and specific degradation pathways of the thawing subsea permafrost. These critical knowledge gaps seriously compromise our ability to assess the current and future contribution of permafrost-derived CH₄ to the global greenhouse gas budget and its feedback on climate.

Wild et al., present unique and thus highly original observations from subsea permafrost drilled on the Laptev Sea shelf. They combine isotopic and biomarker analysis with 10-months incubation experiments to investigate the sources and degradation state of permafrost organic matter and its susceptibility towards microbial decay. They thus directly address one of these critical knowledge gaps and provide novel and important quantitative information that is desperately needed to evaluate the impact of current and projected subsea permafrost thaw on global climate evolution. Therefore, the manuscript makes an important step towards a better quantification of subsea permafrost-climate feedbacks. It provides noteworthy results that are of significance not only to the specific field, but also to the wider carbon cycle/climate community. The methodology looks overall sound and is described in sufficient detail. Please find below detailed comments concerning data analysis, interpretation, and conclusions:

Response: We thank also this reviewer for positive feedback and appreciation of this study of carbon in subsea permafrost. All comments are addressed below in detail.

Environmental settings:

p.7, l. 171: How does the reported permafrost thawing rate of 14 cm/yr compare to other estimates of subsea permafrost thaw on the Siberian Shelf?

How representative are local environmental conditions for permafrost-bearing sediments on the Arctic shelf? In other words, how transferable might results be?

Response: Direct measurements of subsea permafrost thawing rates are extremely difficult, and we are not aware of any other estimate than these 14 cm yr⁻¹ reported by our earlier companion paper Shakhova et al.³. Some indirect information about subsea permafrost thaw is provided by comparing the depth of the permafrost table with estimated inundation time. This comparison supports the thaw rates reported by Shakhova et al., and suggests generally higher thawing rates in the Laptev than in the Beaufort Sea⁴. We added a comment on the lack of other direct observations to the Introduction (lines 71-72).

Sources of OM

L.160-162. Bulk OC contents are low and authors thus assume that OM is derived from fluvial/alluvial sediment deposits. This seems to be confirmed by $\delta^{13}\text{C}$ values (L.186-187), which range between -25.6 – -24.3 ‰. However, one could argue that the observed $\delta^{13}\text{C}$ range still falls within the range of terrestrial OM(?!). Could one not assume that the difference between Ice Complex Deposits and Thermokarsts ($\delta^{13}\text{C} < -26\text{‰}$) and the observed subsea permafrost samples be caused by alteration during transport (eg via rivers) and not necessarily distinct sources?

Response: We fully agree. We clarify in the updated manuscript that the difference between Ice Complex/thermokarst deposits and fluvial deposits is driven by organic matter processing during transport, plausibly amended by mixing with ¹³C-enriched organic matter from freshwater aquatic primary production. We also rephrased to “Average $\delta^{13}\text{C}$ values ... were in line with organic matter modification during fluvial translocation” (lines 176-177).

Then in L.205-207, the authors conclude that the TAR of 32 ± 16 indicates the dominance of OM from lower plants such algae and mosses, thus supporting the statement on L.193-196. However, this seems to be a bit of an oversimplified interpretation of (n-alkanes) $\text{TAR} = \frac{\text{C27}+\text{C29}+\text{C31}}{\text{C15}+\text{C17}+\text{C19}}$, HMW homologs from (higher) plant waxes vs. LMW from aquatic sources. Looking at Table S9, most of C27-C29-C31 values greatly outweigh those of LMW, thus contradicting an aquatic predominance.

Response: We revisited our interpretation of TAR values and agree with the reviewer. While TAR values observed in subsea permafrost (32 ± 16) are considerably lower than those in Ice Complex (420 ± 246) and thermokarst deposits (199 ± 187), they still indicate a dominance of terrigenous over aquatic n-alkanes. The same is true for marine surface sediments in the Buor-Khaya Bay (19 ± 3), where high input of organic matter from the Lena river also leads to $\text{TAR} > 1$, indicating predominance of terrigenous n-alkanes. We revised the text accordingly (lines 188-199).

Then the ration C25/C25+C29 is used to show the relative contributions of Sphagnum mosses (peat-forming) to OC pool (L.209 onwards). The low ratio values do support the low contribution of Sphagnum relative to leaf waxes (C29), but again I do not really see how this would indicate the predominance of aquatic sources (L.210).

Response: We apologize for the confusion; we re-phrased this section of the text (lines 200-208). The *Sphagnum* proxy is used together with lignin phenol ratios to assess the contribution of peat, forest and tundra material to subsea permafrost, and TAR together with ^{13}C is interpreted to indicate modification during aquatic transport.

In summary, the interpretation of n-alkane TAR would benefit from some clarification and a more inclusive discussion of what the actual values may indicate. Lipids represent a relatively minor pool of bulk OM. The TAR values presented here support a shift from TAR values found in ICD and thermokarst to those determined in the subsea permafrost, which may reflect modified during transport (is that what you mean by aquatic translocation?), rather than a predominance of aquatic sources.

Response: Yes, this is what we mean! We clarified the text accordingly (lines 188-199); see also above for details on ^{13}C and lipid proxy interpretation.

Greenhouse gases production

Is there any link between CH₄ (and CO₂) production and OM composition in these incubations? Could that explain the differences in rates and timings of initial production (L.284-286)?

Response: Thanks for the suggestion! We revisited our data and found that CH₄ production (normalized by organic carbon) was highest in samples with a stronger influence of aquatic translocation, indicated by lower organic carbon and nitrogen content, organic carbon over nitrogen ratios and TAR, S/V and C/V lignin phenol proxies suggesting forest sources, higher $\delta^{13}\text{C}$ values and C₂₅/(C₂₅+C₂₉) n-alkane *Sphagnum* proxies. We now added this to the manuscript (lines 311-315).

Could the sediment and/or pore fluid compositions offer further explanations for the comparably low CH₄ production rates and the differences in CO₂/CH₄ production (L.289)? Upon sediment thaw during incubation, some of these compositional features (e.g., background sulfate, nitrate, or iron hydroxides) could inhibit CH₄ production or consume produced CH₄?

Response: We agree that pore fluid composition could have an effect on CH₄ production. We do not have the data to test this, but now added pore fluid composition to the discussion on possible sources of variability (lines 315-319). The potential effect of downward-moving ions such as sulphate *in-situ* is also discussed now in lines 328-333.

Also, the result that subsea permafrost CH₄ production rates are lower than in terrestrial permafrost incubations is an important and novel finding. Would it be possible to identify potential reasons? For instance, what role do thermodynamic limitations (other than temperature), microbial biomass dynamics (also activation from the resting stage), microbial community composition, etc play? Would you expect results to be transferable to other subsea permafrost settings?

Response: We substantially revised the comparison of greenhouse gas production in different deposits, following also the suggestion below to add decomposition models. We now (1) specifically compare greenhouse gas production during one year of incubation, and thereby remove bias on interpretation caused by highly variable length of incubation experiments, and (2) resolve greenhouse gas production in different types of terrestrial permafrost deposits. The now refined comparison shows that CH₄ production in subsea permafrost was similar to that of terrestrial Pleistocene permafrost, but lower than that of terrestrial Holocene permafrost and thermokarst deposits. Identifying the reasons for this remains speculative, given the low number of samples, but we now discuss possible sources of variability in the updated discussion. See lines 315-319, and the new Fig. 7 (the previous figure has been moved to the Supplement).

Wider Impact

The authors should discuss the potential wider impact of their findings in more detail and more comprehensively.

For instance, I would like to strongly encourage the authors to provide a quantitative measure (e.g. first-order rate constant/turn-over time, multi-pool model, reactive continuum) of the subsea permafrost reactivity. This would not only allow a straightforward comparison of the results with the wide range of published OC reactivity data from different settings, but could also inform the set-up and parametrization of regional models. In addition, it would be interesting to compare the observed methanogenesis rates with observations from other extreme environments (e.g. subglacial environments, the deep biosphere).

Response: We appreciate the suggestion and revised our comparison of greenhouse gas production with previous studies. First, we added parameterization of two-pool models for both CO₂ and CH₄ production (lines 291-298). Since previous studies have reported very different developments of CO₂ and CH₄ production over time that cannot be described by one type of model, we were however not able to compare our model fitting parameters with those of previous studies. Instead, we compiled data from previous permafrost incubation experiments lasting at least one year, and compare observed organic carbon loss and CH₄ production during the first year with our data. This observation-based approach is unbiased by interactions between different lengths of incubation experiments and changes in greenhouse gas production rates over time (lines 299-307). We also appreciate the suggestion to compare our data with subglacial deposits and added this now to lines 309-311.

If permafrost OC degradation rates are low, seawater sulphate, diffusing downcore from the sediment-water interface might penetrate into the thawed layer and inhibit methanogenesis, thus limiting CH₄ production.

Response: Previous observations indeed support the diffusion of sulphate into subsea permafrost after thaw, down to the permafrost table^{5,6}. However, only one of the incubated samples was taken above the permafrost table (and did not show the lowest CH₄ production rates), so it is unlikely that this was a main driver of variability in our incubations. We however now added pore fluid composition in general as a possible driver of variability between samples and compared to terrestrial permafrost deposits as suggested above (lines 315-319), and now discuss also the possibility of downward-diffusing sulphate to inhibit methanogenesis (lines 328-333).

The authors also compare measured CH₄ production rates with ocean CH₄ emissions. However, a plethora of past studies have shown that, generally, strong CH₄ sinks in both ocean and sediment efficiently consume a large fraction of deep CH₄ fluxes. Thus, one would expect production rates to be higher than observed CH₄ emissions at the ocean surface.

Response: We agree that methane oxidation might filter a substantial fraction of produced methane before it reaches the ocean surface (although quantitative data from the East Siberian Arctic Ocean are missing). We re-structured our discussion of methane fluxes and oxidation to clarify this point (lines 335-337), and specifically emphasize CH₄ oxidation when comparing with ocean emissions: “*The lower rates of CH₄ production by subsea permafrost decomposition estimated here, and the likely oxidation of part of this CH₄, do not point at a dominant contribution of organic matter decomposition in thawed subsea permafrost to the high emissions observed in the study area. ... Taken together, the high methane emissions fluxes ubiquitously observed in the field likely stem from other sources such as from preformed methane in gas pockets in the subsea permafrost, collapsing CH₄ hydrates or venting of a deep thermogenic CH₄ pool.*” (lines 339-346)

The influence of thawing subsea permafrost on the ocean’s carbonate system and saturation state is somewhat (over?)simplified. It will depend on the complex and dynamic interplay of different early diagenetic processes. While aerobic degradation of OC (for instance in the water column or the shallow sediment layers) would indeed result in the acidification of overlaying waters, the anaerobic degradation of OC in deeper sediment layers and the multitude of strongly coupled redox processes make it difficult to predict the possible impact that will depend on the delicate balance between

aerobic and (an)aerobic CH₄ oxidation processes, as well as a number of mineral precipitation/dissolution processes.

Response: We agree and now address factors influencing CO₂ release to the water column in lines 363-366: “How much of the CO₂ produced by subsea permafrost decomposition is eventually released to the water will however depend on CO₂ consumption by microbial processes and the balance of precipitation and dissolution reactions.”

References

1. Abramov, A. *et al.* Two decades of active layer thickness monitoring in northeastern Asia. *Polar Geogr.* 1–17 (2019).
2. CALM. Circumpolar Active Layer Monitoring (CALM) program database. (2021).
3. Shakhova, N. *et al.* Current rates and mechanisms of subsea permafrost degradation in the East Siberian Arctic Shelf. *Nat. Commun.* **8**, 15872 (2017).
4. Angelopoulos, M., Overduin, P. P., Miesner, F., Grigoriev, M. N. & Vasiliev, A. A. Recent advances in the study of Arctic submarine permafrost. *Permafr. Periglac. Process.* **31**, 442–453 (2020).
5. Ulyantsev, A. S., Polyakova, N. V., Romankevich, E. A., Semiletov, I. P. & Sergienko, V. I. Ionic composition of pore water in shallow shelf deposits of the Laptev Sea. *Dokl. Earth Sci.* **467**, 308–313 (2016).
6. Overduin, P. P. *et al.* Methane oxidation following submarine permafrost degradation: Measurements from a central Laptev Sea shelf borehole. *J. Geophys. Res. Biogeosciences* **120**, 965–978 (2015).

Reviewers' Comments:

Reviewer #2:

Remarks to the Author:

I have now carefully read the author's response to the review, as well as the revised manuscript. The authors have provided a very careful, thoughtful, and detailed response to the points raised and have modified the manuscript accordingly. I really appreciate the authors' openness towards the suggested modifications/additions and their efforts to address them. The raised concerns. This has been the most constructive and pleasant review experience I had in a very long time. The modifications and additions have significantly improved the manuscript and I am thus happy to recommend the publication of these exciting new findings.

Nonetheless, I would still like to ask the authors to consider clarifying the following minor points before publication:

2 Pool Degradation Model

In general, the degradability of the bulk OM is controlled by the different abilities of a large number of microbes that compete for a common substrate composed of a complex mixture of different compounds. As a consequence, the degradability of the bulk OM is continuously (and probably also dynamically) distributed over a large range of OM degradability. Such a continuous distribution can be approximated by a discrete representation over well-defined time scales (i.e. like an integral can be approximated by a discrete sum). However, such approximation also results in limitations that should be acknowledged:

- 1) P.6, line 294: Consider reformulating to reflect that the good model-data fit indicates that OM degradation dynamics over the considered incubation period can be well approximated by considering two, discrete OM pools: a very small (0.001% of initial OM), easily degradable ($k=??$) pool and a larger (>99% of initial OM), less degradable pool ($k=??$).
- 2) P.13, line 327: I would suggest adding a statement about the predictive ability of the model. The 2 pool approximation provides a good fit to the degradation dynamics of the considered incubation period. However, the 2 pool model converges to a constant degradability once the easily degradable pool is depleted. In the real world, OM degradation tends to continuously decrease with time across a wide range of different environments. You might thus overpredict CH₄ production beyond the incubation timescale.

CO₂ released into the water column

p. 14, line 364: Consider clarifying that it is not only uncertain how much dissolved carbon (DIC=CO₂+HCO₃+CO₃) escapes the sediment, but also in which form. The speciation of DIC is controlled by ambient pH, which in turn is influenced by almost every single diagenetic process that takes place in the sediment. The exact speciation of this benthic DIC flux is important for assessing its impact on the Arctic Ocean ecosystem.

Organic matter composition and greenhouse gas production of thawing subsea permafrost in the Laptev Sea

Birgit Wild, Natalia Shakhova, Oleg Dudarev, Alexey Ruban, Denis Kosmach, Vladimir Tumskey, Tommaso Tesi, Hanna Joss, Inna Nybom, Felipe Matsubara, Helena Alexanderson, Martin Jakobsson, Alexey Mazurov, Igor Semiletov, Örjan Gustafsson

We are delighted about the positive news and address below the last few minor points. Reviewer comments are presented in blue italics, with author responses in black normal font.

Reviewer #2 (Remarks to the Author):

I have now carefully read the author's response to the review, as well as the revised manuscript. The authors have provided a very careful, thoughtful, and detailed response to the points raised and have modified the manuscript accordingly. I really appreciate the authors' openness towards the suggested modifications/additions and their efforts to address them. The raised concerns. This has been the most constructive and pleasant review experience I had in a very long time. The modifications and additions have significantly improved the manuscript and I am thus happy to recommend the publication of these exciting new findings.

Response: Thanks a lot! We appreciate the suggestions in this and the previous round of review, and we think that integrating these changes has really improved the manuscript. We address the last three points below.

Nonetheless, I would still like to ask the authors to consider clarifying the following minor points before publication:

2 Pool Degradation Model

In general, the degradability of the bulk OM is controlled by the different abilities of a large number of microbes that compete for a common substrate composed of a complex mixture of different compounds. As a consequence, the degradability of the bulk OM is continuously (and probably also dynamically) distributed over a large range of OM degradability. Such a continuous distribution can be approximated by a discrete representation over well-defined time scales (i.e. like an integral can be approximated by a discrete sum).

However, such approximation also results in limitations that should be acknowledged:

1) P.6, line 294: Consider reformulating to reflect that the good model-data fit indicates that OM degradation dynamics over the considered incubation period can be well approximated by considering two, discrete OM pools: a very small (0.001% of initial OM), easily degradable ($k=??$) pool and a larger (>99% of initial OM), less degradable pool ($k=??$).

Response: We fully agree on the limitations of 2-pool models in reflecting the continuity of OM degradability. We included a statement on the interpretation of the model fit as suggested, describe pool sizes in the text and refer to the Supplement for details on fitted parameters (including k values; lines 296-297).

2) P.13, line 327: I would suggest adding a statement about the predictive ability of the model. The 2 pool approximation provides a good fit to the degradation dynamics of the considered incubation period. However, the 2 pool model converges to a constant degradability once the easily degradable pool is depleted. In the real world, OM degradation tends to continuously decrease with time across a wide range of different environments. You might thus overpredict CH_4 production beyond the incubation timescale.

Response: We also agree on this point and added a statement on the constant CH_4 production rate at the end of the incubation and the resulting likely overestimation of CH_4 production beyond the incubation timescale (lines 333-336).

CO₂ released into the water column

p. 14, line 364: Consider clarifying that it is not only uncertain how much dissolved carbon ($DIC=CO_2+HCO_3+CO_3$) escapes the sediment, but also in which form. The speciation of DIC is controlled by ambient pH, which in turn is influenced by almost every single diagenetic process that takes place in the sediment. The exact speciation of this benthic DIC flux is important for assessing its impact on the Arctic Ocean ecosystem.

Response: We agree and added a sentence on DIC speciation as suggested (lines 371-373).